# Use of Defensins to Develop Eco-Friendly Alternatives to Synthetic Fungicides to Control Phytopathogenic Fungi and Their Mycotoxins

**DOI:** 10.3390/jof8030229

**Published:** 2022-02-25

**Authors:** Valentin Leannec-Rialland, Vessela Atanasova, Sylvain Chereau, Miray Tonk-Rügen, Alejandro Cabezas-Cruz, Florence Richard-Forget

**Affiliations:** 1Université de Bordeaux, UR1264 Mycology and Food Safety (MycSA), INRAE, 33882 Villenave d’Ornon, France; valentin.leannec-rialland@inrae.fr; 2UR1264 Mycology and Food Safety (MycSA), INRAE, 33882 Villenave d’Ornon, France; vessela.atanasova@inrae.fr (V.A.); sylvain.chereau@inrae.fr (S.C.); 3Institute for Insect Biotechnology, Justus Liebig University, Heinrich-Buff-Ring 26-32, 35392 Giessen, Germany; miray.tonk@agrar.uni-giessen.de; 4Institute of Nutritional Sciences, Justus Liebig University, Wilhelmstrasse 20, 35392 Giessen, Germany; 5Anses, Ecole Nationale Vétérinaire d’Alfort, UMR Parasitic Molecular Biology and Immunology (BIPAR), Laboratoire de Santé Animale, INRAE, 94700 Maison-Alfort, France

**Keywords:** plant disease, fungal pathogens, mycotoxins, biocontrol strategies, defensins

## Abstract

Crops are threatened by numerous fungal diseases that can adversely affect the availability and quality of agricultural commodities. In addition, some of these fungal phytopathogens have the capacity to produce mycotoxins that pose a serious health threat to humans and livestock. To facilitate the transition towards sustainable environmentally friendly agriculture, there is an urgent need to develop innovative methods allowing a reduced use of synthetic fungicides while guaranteeing optimal yields and the safety of the harvests. Several defensins have been reported to display antifungal and even—despite being under-studied—antimycotoxin activities and could be promising natural molecules for the development of control strategies. This review analyses pioneering and recent work addressing the bioactivity of defensins towards fungal phytopathogens; the details of approximately 100 active defensins and defensin-like peptides occurring in plants, mammals, fungi and invertebrates are listed. Moreover, the multi-faceted mechanism of action employed by defensins, the opportunity to optimize large-scale production procedures such as their solubility, stability and toxicity to plants and mammals are discussed. Overall, the knowledge gathered within the present review strongly supports the bright future held by defensin-based plant protection solutions while pointing out the obstacles that still need to be overcome to translate defensin-based in vitro research findings into commercial products.

## 1. Introduction

Fungal plant diseases jeopardize global food security. Actually, staple crops with high economical and agronomical value, including rice, wheat, maize, potato and soybean, are threatened by various fungal diseases that can lead to substantial yield losses [1,2,3]. Using the harvest statistics provided by the Food and Agriculture Organization (FAO) for the period 2009–2010, Fisher et al. [4] estimated that the losses caused by fungal diseases with regard to wheat, rice, maize potato and soybean were equivalent to the food necessary to feed 600 million humans over one year. Of greater concern is that the currently available knowledge does not allow ruling out the possibility that climate change would increase the impact of major fungal plant diseases as well as create environmental conditions promoting the emergence of new devastating fungal diseases [5,6]. Among the phytopathogenic fungal species recognized as the most economically important, one can mention *Magnaporthe oryzae*, which is responsible for rice blast that can lead to up to 35% harvest losses; *Botrytis cinerea*, which causes severe damages to a broad range of plant species; *Puccinia* species and the two *Fusarium* species, *Fusarium graminearum* and *Fusarium oxysporum*, which cause significant damages to diverse crops [7]. In addition to jeopardizing crop yields, some phytopathogenic fungi can also significantly affect crop safety as a result of their capacity to produce mycotoxins. This is notably the case of several *Fusarium* species infecting cereal crops [8] but also of various species within the *Aspergillus*, *Penicillium*, *Alternaria*, and *Claviceps* genera that can contaminate a wide variety of agricultural products [9]. Mycotoxins are fungal secondary metabolites causing serious adverse health effects to both humans and livestock [10]. The most important mycotoxins that affect health and agro-economy are aflatoxins, patulin, trichothecenes, zearalenone, fumonisins, ochratoxin A and ergot alkaloids [11]. According to FAO estimates, 25% of the world’s crops are contaminated by mycotoxins above the limits set by national and agricultural regulations, which leads to annual losses close to 1 billion metric tons [10]. These estimates were recently refined in the report of Eskola et al. [12] which indicates that 60–80% of agricultural products contain detectable levels of mycotoxins. Despite increasing efforts to develop agronomic and cultural practices to manage and control plant infecting fungi including crop rotation, and to improve appropriate management of crop residues and the appropriate use of resistant cultivar when available [13,14], the application of synthetic fungicides has been the primary strategy adopted by farmers and is still widely used. However, concerns over environmental contamination and human health risks [15], restrictions or cancellations of authorization by some countries, have driven research to develop safe and efficient alternatives to synthetic fungicides. To prevent the emergence of resistant fungal strains, as has been observed with the intensive use of single-target site fungicides, priority should be given to multi-target solutions [16,17,18]. Based on their nature, control methods’ alternatives to conventional fungicides can be classified as chemical or biological. Biological solutions include the use of plant growth-promoting bacteria, mycorrhizal fungi to promote plant fortification and/or enhance plant defense, and the use of antagonists microorganisms that are able to counteract the spread of the fungal pathogen [19,20,21]. Chemical solutions leverage the capacity of molecules from natural origin to prevent or reduce fungal growth. Among natural molecules, antimicrobial peptides (AMPs), that can be produced either by animals, plants or fungi, have been the subject of increasing research in recent decades. AMPs are low molecular mass biomolecules, generally between 12 and 50 amino acids, that play an important role in innate host defense against microbial colonization [22] and possess a wide range of antimicrobial activities against bacteria, fungi, viruses and protozoa [23]. According to the presence of α-helix and/or β-sheet secondary structural elements, AMPs are commonly divided into four categories represented in Figure 1; α, β, αβ and non-αβ [24]. For instance, the human cathelicidin LL37, which has been widely studied due to its large repertoire of functional activities including direct antimicrobial activities against various types of microorganisms, belongs to the α-helical peptide category of AMPs [25]. As an example of β-sheet peptides, one can mention gomesin, which has been isolated from the spider *Acanthoscurria gomesiana* and contains two β-sheets linked through two disulfide bridges forming a β-hairpin motif [26]. Indolicidin, a 13-amino-acid-long peptide with a linear structure isolated from bovine neutrophils is a typical example of the class of non-αβ AMPS [27]. Regarding the mixed αβ category, this class includes, but is not limited to, several microcins such as the microcin B17 that is produced by strains of *Escherichia coli* and displays efficient bactericidal activity [28]. Of all the AMPs reported thus far, the defensin family which comprises peptides with α-helix and/or β-sheet has been the most extensively studied. Their antimicrobial activity has been evidenced against a broad range of human and plant pathogens, including bacteria, oomycetes, virus, fungi or even apicomplexan parasites [29,30,31,32,33,34]. One specific defensin, such as MtDef4 or MtDef5 from *Medicago truncatula*, can exhibit a wide antimicrobial spectrum and can be active against both human and plant pathogens while others (such as the D2 defensin from *Spinacia oleracea*) exhibit a more narrow spectrum of activity [35]; additionally, one pathogen can be affected by different defensins [32,34,36]. Interestingly, some defensins have been shown to display antifungal activity against pathogens, leading to devastating disease in crops. For instance, the defensins RsAFP2 from *Raphanus sativus* and Sa-AFP2 from *Sinapis alba* have been demonstrated as potent inhibitors of the fungal growth of major pathogens including *Fusarium culmorum* and *B. cinerea* [29]. This has led several authors to propose defensins as promising candidates for medical and agricultural applications, including the treatment of life-threatening microbial diseases or treatments against phytopathogens [37,38,39].

In this review, we will specifically focus on the potential use of defensins as novel leads for the development of sustainable solutions to control plant fungal diseases, reduce yield losses and mycotoxin contamination and therefore improve food security and safety. Firstly, the most recent information on the major characteristics of defensins, their antifungal activity and mechanisms of action will be discussed. Secondly, the biological applications of plant defensins as eco-friendly alternatives to synthetic fungicides will be debated.

## 2. Origin and Characteristics of Defensins

The term defensin was introduced in 1985 to refer to peptides with antimicrobial activities isolated from humans [40]. Since then, the term defensin has been expanded to include peptides from non-human organisms possessing functional (antimicrobial properties) and structural (a compact cysteine-stabilized β-sheet structure) similarities. Defensins constitute the largest, and most studied, group of AMPs [41]. These small proteins of approximately 20–60 amino acids are ubiquitous and multipotent components of the innate immune system of a wide range of organisms within the animal, plant and fungi kingdoms [42,43]. The defensins are cationic cysteine-rich peptides with a high diversity of amino acid sequence. However, despite this low level of amino acid sequence identity, most defensins bear some similarities in their tertiary structure stabilized into compact shapes [44].

Defensins are separated into two principal super-families, the cis- and trans- defensins, with an independent evolutionary origin and a convergent evolution of their structural folds [44,45]. The cis- and trans-classification is based on the spacing and pairing of the cysteine residues and the orientation of the peptide’s secondary structure. Defensins of the cis-family contain two parallel cis-oriented disulfide bridges pointing in the same direction and stabilizing the same β-strand to an α-helix. Cis-defensins have been reported in a wide array of invertebrate animals, fungi and spermatophyte plants [44]. The cis-defensins have generally more diverse and longer amino acid sequences than trans-defensins. The trans-family of defensins is characterized by two trans-oriented disulfide bridges pointing in opposite directions from the final β-strand and thus stabilizing different structural elements. Trans-defensins have either been observed in invertebrates or vertebrates [45] and include α-defensins, β-defensins and big defensins, this last family being supposed to be the ancestors of β-defensins [46]. In addition to these families, it is worth mentioning the occurrence of θ-defensins, which are the only cyclic peptides of animal origin reported to date [47].

The tertiary structure of defensins is characterized by the connectivity pattern of their disulfide bridges, which is unique to their phylum and conserved within the defensin family as represented in Figure 2 [44]. Thus, all vertebrate α-defensins have three disulfide bridges between cysteine (Cys) residues, Cys1–Cys6, Cys2–Cys4, and Cys3–Cys5. In vertebrate β-defensins, the three-disulfide bridges are between Cys1–Cys5, Cys2–Cys4, and finally Cys3–Cys6. For cis-oriented defensins in invertebrates, the common linkage pattern is Cys1–Cys4, Cys2–Cys5, Cys3–Cys6. In plant defensins, the disulfide bonds between cysteine residues commonly share the same following pattern, Cys1–Cys8, Cys2–Cys5, Cys3–Cys6, and Cys4-Cys7 [42]. In plant and some invertebrate (notably arthropods and mussels) defensins, disulfide bridges connect one α-helix and three or two-strand antiparallel β sheets leading to a stabilized motif called cysteine-stabilized alpha-beta (CSαβ) schematized in Figure 3 [48]. Defensins containing a CSαβ motif, also designed as CSαβ-defensins, have been categorized in three major types, namely antibacterial ancient invertebrate-type defensins (AITDs), antibacterial classical insect-type defensins (CITDs) and antifungal plant/insect-type defensins (PITDs) [49]. In contrast to plant and invertebrate defensins, mammal defensins usually do not contain α-helices and consequently no CSαβ motif [50]. The presence of disulfide bridges in the defensin structure confers a high stability against chemical and thermal extreme conditions to this class of peptides [51,52], such as protection from cleavage by proteolysis [53]. Defensins often adopt an amphipathic structure with a hydrophobic side facing a hydrophilic one, which, in addition to their typically cationic state (net charge inter-quartile range from +1 to +5), facilitates the interaction and insertion of the peptides into the anionic cell walls and the double layer of phospholipid membranes of microorganisms [54]. Defensins possess a structural residue characterized as functionally important located in the C-terminal β-sheet domain. This motif, conserved across all classes of CSαβ-defensins, is called γ-core. The γ-core is assumed to be responsible for the antimicrobial activity of defensins as it has been demonstrated for several plant defensins including RsAFP2, Psd1, MsDef1, and MtDef4 [55,56,57], but also of metazoan defensins such as tick defensins [36].

Defensins are synthetized as precursor proteins that possess an N-terminal endoplasmic reticulum targeting signal peptide followed by the mature defensin domain and an optional C-terminal prodomain [58,59]. According to the presence or absence of the C-terminal prodomain, plant defensins are divided into two classes: class I (absence of the C-terminal prodomain) and class II (presence of the C-terminal prodomain). The role of the C-terminal prodomain in the *N. alata* NaD1 defensin was investigated by Lay et al. [60]. The previous authors have shown that this pro-peptide which is reach in hydrophobic and acidic amino acids is involved in targeting the vacuoles and eliminating the potential detrimental effects caused by the basic nature of the defensin in the plant host cells [60].

## 3. Activity of Defensins against Fungal Phytopathogens

While the bactericidal activity of defensins has been extensively characterized, their antifungal activity has been relatively less studied [61]. In addition, most of the available scientific literature refers to the activity of defensins against human fungal pathogens and there is much less information regarding their capacity to inhibit the growth of fungal plant pathogens. The available information regarding defensins and defensin-like peptides (DLP) reported as active against economically important plant-infecting fungi including mycotoxigenic fungi (e.g., *Fusarium* sp., *Penicillium* sp., *Aspergillus* sp., *Alternaria* sp.) is presented in Table 1.

As shown in Table 1, most defensins that have been characterized to date for their capacity to restrain the growth of plant-infecting fungi belong to the plant defensin group. Among the 67 plant defensins and DLPs identified through our literature search, the majority of them were isolated from plants of the *Fabaceae* (mainly related to various *Medicago*, *Vigna* and *Pisum* species) and *Brassicacea* (e.g., *Raphanus*, *Sinapis*, *Arabidopsis* and *Brassica* species) families. RsAFP1 and RsAFP2 from *Raphanus* and *Brassica* species [29,63,67], MtDef2 and MtDef4 from *M. truncatula* [35,56] and Nad1 and Nad2 from *N. alata* [98,99] were those for which the antifungal activity against plant pathogens were the most extensively documented. Regarding invertebrate defensins, 22 peptides have been shown as efficient to restrain the growth of plant infecting fungi. With the exception of Cg-Def isolated from *C. gigas* [109] and MGD-1 from *M. galloprovincialis* [110], these antifungal invertebrate defensins have been found in insect and arachnid species. The literature review highlighted six defensins from filamentous fungi and six defensins from vertebrates with a reported activity against phytopathogenic fungi: three occurring in fish species, one from a snake species, one in a penguin species and one homologue of the Drosophila-derived drosomycin observed in humans [114,115,116,117,118]. The small proportion of fungi and animal defensins listed in Table 1 supports previously published conclusions indicating that plant defensins primarily exhibited activity against fungi while fungal and animal defensins have efficient antibacterial properties [48]. The previous statement should, however, be put in balance with the history of research dedicated to defensins. Actually, as illustrated in the review of Silva et al. [41], the research addressing the antifungal bioactivity of defensins has only increased in a really more recent past that dedicated to antibacterial effects. It is therefore reasonable to assume that the antifungal activity of animal defensins, which were the first identified defensins, has been understudied. To evidence the antifungal properties of defensins, a broad set of targeted fungi has been used. The list reported in Table 1 includes fungi responsible for major plant diseases, such as the phytopathogenic fungi of cereal crops (*F. culmorum* and *F. graminearum*, *S. tritici* and *Pycularia oryzae*) or of cruciferous crops (*Leptosphaeria maculans*), fungi affecting grape quality (*B. cinerea*) and fungi infecting fruit and vegetable crops (*F. oxysporum*, *F. solani*, *N. haematococca*). Among this list of targeted fungi, several species are acknowledged as responsible for crop contamination by mycotoxins. This is the case, for example, of *F. graminearum* and *F. culmorum* that are the main causal agents of cereal contamination with deoxynivalenol mycotoxin [8] of *F. verticillioides* that produces fumonisins on maize grains [124] and of the ochratoxin-producing *A. niger* species and the sterigmatocystin-producing *A. versicolor* and *nidulans* species [125].

To assess the antifungal efficacy of defensins, MIC and/or IC_50_ were used (Table 1). It should be borne in mind that the heterogeneity of experimental procedures targeting the fungal strain and fungal growth assessment method—especially with regard to culture conditions—makes it difficult to compare results from different studies. Nevertheless, for the tests realized within a same study and using similar protocols, differences in MIC and IC_50_ values may reveal the occurrence of variations in the specificity of defensins towards pathogens and/or in their mode of action. Thus, the data reported in Table 1 indicate a significantly higher efficacy of the AFP defensin from *A. giganteus* against *F. sporotrichioides* (MIC value of 0.1 µg/mL which corresponds to a 0.02 µM concentration) than against *F. culmorum* (MIC value higher than 70 µM) [119]. Such differences in antifungal efficacy were also reported for the RsAFP1 defensin from *R. sativus* that has been characterized by a 0.05 µM MIC value when tested against the rice blast fungus *P. oryzae* and a 17.6 µM MIC value against the Basidiomycota *R. solani* [67]. The RsAFP1 defensin was also shown to be twice as efficient against *F. oxysporum f.* sp. *Pisi* (IC50 = 2.65 µM) than against *F. oxysporum f.* sp. *Lycopersici* (IC50 = 5.3 µM) [29,67]. Additionally, as illustrated with PAF and PAFB from *P. chrysogenum* tested against various *Aspergillus* species [120], different defensins from the same origin can display important disparity in their antifungal effectiveness and their target specificity. Finally, data gathered in Table 1 also support the point that one fungal species can be more or less affected by defensins of different origin. Thus, *B. cinerea* was shown to be approximately twice as sensitive to the DM-AMP1 defensin from *D. merckii* than to the Ah-AMP1 defensin from horse chestnut *A. hippocastanum* [63].

In addition to assessing the antifungal efficacy of defensins, some authors have considered the specific activity of their γ-core. For instance, Tonk et al. [36] have reported that the γ-core of the defensin DefMT3 was two to four times more efficient in inhibiting the spore germination of *F. graminearum* and *F. culmorum* than the mature defensin. In contrast, the γ-core of the defensins MtDef4 and MtDef5 exhibited a lower inhibitory potential against Ascomycota *F. oxysporum* and *P. medicaginis* than the parental defensins [35]. These opposite results may be related to the absence/presence of disulfide bridges and/or creation of oligomers. To identify the determinants of the γ-core activity, structure/function investigations have been implemented; and γ-core sequences and degree of inhibitory efficiency have been compared. Such approaches have allowed Lacerda et al. [126] and Leannec-Rialland et al. [127] to demonstrate that the positively charged amino acids located in the γ-core were essential for the antifungal activity; other structural motifs responsible for antimicrobial activity being the α-patch, the γ-patch, and m-loop [128,129].

Although several reports have documented the antifungal activity of defensins against plant pathogens, very few have investigated their potential to inhibit the yield of mycotoxins. To our knowledge, this potential was first demonstrated by Leannec-Rialland et al. [127], who showed the remarkable efficacy of the γ-core of the tick defensin DefMT3 to inhibit the production of type B trichothecenes by *F. graminearum*. The previous authors also evidenced that the tertiary structure of the peptide, the occurrence of dimer forms and its cationic properties were primary factors involved in the mycotoxin inhibition activity of DefMT3 γ-core.

## 4. Antifungal Mechanism of Action of Defensins

Defensins with an acknowledged antifungal activity are classified into two groups according to their antimicrobial action: (i) the morphogenic defensins, causing a reduced hyphal elongation with an increase in hyphal branching; and (ii) the non-morphogenic defensins that lead to a reduction in the hyphal elongation without provoking observable changes in hyphae morphology [80,130,131]. For instance, MsDef1 from *M. sativa* that induces the important hyperbranching of fungal hyphae belongs to morphogenic group while MtDef4, from *M. truncatula*, is non-morphogenic [132]. A variety of key features have been proposed to explain the antifungal activity of defensins. These features, schematized in Figure 4 and detailed in the following, are related to fungal membrane binding and the induction of membrane disorders, as well as to the production of reactive oxygen species (ROS) and their interaction with fungal specific targets once the defensin has entered the cytoplasm. According to this multifaceted mechanism of action, defensins have been shown to affect various fungal pathways. In the recent publication of Aumer et al. [133], the use of a proteomic approach has allowed evidencing the alteration of spliceosome, ribosome protein processing in endoplasmic reticulum, endocytosis, MAPK signaling pathway and oxidative phosphorylation in *B. cinerea* exposed to an analogue of the insect defensin heliomicin. In addition to being comprehensively reviewed by Parisi et al. [134] and Struyfs et al. [135], the antifungal activity of different defensins can result from different mechanisms. While some defensins require crossing the fungal cell wall and plasma membrane to induce cell death, others can exert their toxic effects from the extracellular side of the fungal cells. Moreover, a single defensin can have different mechanisms of action depending on the targeted fungal species [136].

### 4.1. Interactions with Host Membrane Components and Induction of Fungal Membranes Disorders

For some defensins, the interaction with specific sphingolipids and phospholipids of the plasma membrane is a prerequisite for their antifungal activity [137]. For example, the binding of the DmAPM1 defensin from *D. merckii* to the sphingolipid mannosyl di(inositolphosphoryl)-ceramide has been shown to be critical for triggering its antifungal activity [138]. The specific target of several defensins including MsDef1 from the barre clover *M. sativa*, Sd5 from the sugarcane *S. officinarum*, RsAFP2 from the radish *R. sativus* and Psd1 from the pea *P. sativum* has been identified as glucosylceramide [139,140,141,142]. MtDef4 from *M. truncatula* has been shown to specifically interact with phosphatidic acid, a precursor of membrane phospholipids and a signaling lipid, and this interaction has been indicated as necessary for MtDef4 entry into fungal cells [143]. Regarding the defensin NaD1 from *N. alata* and the tomato defensin TPP3, their interaction with phosphatidylinositol (4,5)-bisphosphate, located in the inner leaflet of the membrane, has been reported as essential for the initiation of their cytotoxic effects [144,145]. More recently, the membrane modeling approach used by Leannec-Rialland et al. [127] indicated that the γ-core of the tick defensin DefMT3 was recruited by the phospholipids POPS, POPA and POPG that are present in the *F. graminearum* membrane. Using the in silico modeling or mutational analysis of amino acids, some specific residues located in the loop region of the γ-core motif, such as Phenyl-alanine 28 and Isoleucine 29 in the DefMet3 protein [32] or the RGFRRR motif in MtDef4 [143], have been predicted as critical for the interaction with the lipid bilayer membrane. The binding site of NaD1 was also characterized: this binding site is formed by the Lysine 4 residue and a KILRR motif located between the β-strands of its γ-core motif [144]. In addition to structural features of the γ-core motif, the specific residues located in loop 1 of some defensins have been demonstrated to be involved in the binding with fungal membranes. For example, the Phenylalanine 15 and the Threonine 16 residues present in Loop 1 of the Psd1 defensin have been shown to be involved in the interaction with glucosylceramide [146].

As a result of the binding with membrane components, defensins can create pores and permeabilize the membranes, which is, however, acknowledged as only one among several mechanisms involved in the antimicrobial action of defensins [147]. This capacity of pore formation is not shared by all defensins; certain defensins such as plectasin—a fungal defensin from *Pseudoplectania nigrella*—does not affect fungal membrane integrity [148]. Actually, neither pore formation, nor changes in membrane potential, nor carboxy-fluorescein efflux from liposomes were detected by the previous authors when *Bacillus subtilis* were exposed to plectasin. The mechanism involved in plectasin bactericidal activity was reported to be associated with an inhibition of membrane-associated steps of cell-wall biosynthesis [148]. The membrane permeabilization of *Neurospora crassa* caused by various plant defensins was reported by Thevissen et al. [149]—the extent of which is dependent on the defensin dose. Such a membrane-permeabilizing activity was also evidenced for NaD1, which was reported to form a relatively stable aperture with an internal diameter ranging between 14 and 23 Å in *F. oxysporum* membrane [150] and for MtDef5 in *F. graminearum* and *N. crassa* [151]. The capacity of defensins to cause membrane permeabilization is dependent on the fungal target as illustrated for MtDef4. Indeed, while MtDef4 has been shown to induce permeabilization in *F. graminearum*, this mechanism did not appear to contribute to the antifungal effect of MtDef4 against *N. crassa* [136]. Some defensins form oligomers and those oligomers were reported as being the active structures associated with membrane permeabilization and antimicrobial activity [152]. This is the case of defensin SPE10, from the plant *P. erosus*, for which the dimeric form was shown to possess high antifungal properties, possibly favored by its increased hydrophobicity [86]. Similarly, the TPP3 tomato defensin can form a dimeric cationic grip through antiparallel alignment of the β strands, stabilized by hydrogen bonds and salt bridge interactions, which was shown as critical for its interaction with PIP2 (phosphatidylinositol 4,5-bisphosphate) and cytolytic activity [145]. NAD1 from *N. alata* was observed to create an arrangement with seven dimers binding to the anionic headgroups of 14 PIP2, leading to a complex oligomer seemingly important for cell permeabilization [144].

There are currently at least three different commonly accepted models describing the possible membrane-permeabilizing activity of defensins: the barrel-stave pore model, the toroidal pore model and the carpet model. To address these specific pore models in greater depth, we strongly encourage the readers to consult the relevant review of Brogden published in 2005 [153]. Briefly, in the barrel-stave model, antifungal peptides self-aggregate in the membrane in a way that their hydrophobic sites face the phospholipid layers of the membrane while their hydrophilic segments face the lumen of transmembrane pores. In the toroidal model, antifungal peptides and membrane lipids interact to form pores that are lined by both peptide and lipid headgroups. In the carpet model, antifungal peptides bind, in a monomeric or oligomeric form, onto the surface of the negatively charged target membrane and surround it in a carpet-like manner, leading to the disruption of the bilayer curvature and the disintegration of the membrane. The immediate consequence of pore-formation induced by some defensins in fungal membranes is the dissipation of ionic gradients and membrane potential across the cytoplasmic membrane of target cells, triggering cell death. The such dysfunction of calcium influx and potassium efflux can also directly result from the binding of defensins with fungal membrane components. In this way, a membrane potential disruption effect has been proposed to explain the activity of a synthetic tick defensin against *Micrococcus luteus* [154]. Similarly, the plectasin fungal defensin [155] and the *Arabidopsis* defensin AtPDF2.3 [65] were proven to interfere with potassium channels. The pea defensin Psd1 was also characterized for its capacity to disturb potassium channels in mammalian cells; however, this activity was not observed in fungal cells [156]. The maize defensin called γ-zethionin was also reported to affect sodium currents by hindering voltage-operated channels [157]. In the same way, MsDef1 was evidenced to perturb calcium exchanges in mammalian cells; a blocking of calcium channels was also supposed to be involved in its antimicrobial action against *F. graminearum* [80]. According to the reports of Zhu et al. [158] and Meng et al. [159], the structural Csαβ-motif could be a key determinant involved in the capacity of defensins or DLPs to block ion channels.

### 4.2. Induction of Oxidative Stress and Apoptosis

There is compelling evidence that defensins can induce ROS accumulation within the targeted fungal cells. This has been notably demonstrated for RsAFP2 in *C. albicans* [160,161], for NaD1 in *C. albicans* [162,163] or in *F. oxysporum* [150] and for HsAFP1 in *C. albicans* [164]. It should be noted that internalization is not required for inducing ROS production as RsAFP2, which is not internalized, induces the production of ROS [161]. ROS can instantaneously and nonspecifically react with essential biological molecules and lead to an alteration of cellular functions by inducing damages such as mutations in DNA, oxidations of proteins, or the peroxidation of lipids. These damages are generally deleterious, and could lead to apoptosis and cell death. The induction of apoptosis in *C. albicans* cells exposed to the OsAFP1, RsAFP2 and HsAFP1 defensins, has been clearly demonstrated thanks to the use of epifluorescence methods [161,164,165]. Regarding the effect of RsAFP2 in *C. albicans*, apoptosis induction was shown to concomitantly occur with an activation of caspases or caspase-like proteases [161]. Since it is strongly suspected that the biosynthesis of mycotoxins could help the fungal cell maintain safe levels of intracellular ROS [166], it makes sense to suggest that ROS accumulation triggered by defensins could affect the production of mycotoxins by toxigenic fungi. However, to date, this potential link between ROS induction by defensins and modulation of mycotoxin yield has not been addressed.

### 4.3. Internalization and Intracellular Targets

Th use of fluorescently labeled peptides coupled to confocal microscopy has boosted the demonstration of cell internalization of various defensins. The translocation of defensins across fungal cell membrane can occur in a non-disruptive manner, frequently for peptide concentrations and/or exposure times that do not lead to significant growth alteration. For instance, while MtDef4 was shown to permeabilize the plasma membrane of *F. graminearum* before its entry into fungal cells, the internalization of MtDef4 into *N. crassa* cells was reported to occur without membrane permeabilization [136]. According to previous works, MtDef4 internalization in *N. crassa* could be related to endocytosis. Similarly, NaD1 has been reported to bind to a putative cell wall receptor of *C. albicans* and to be taken up to the cytoplasm through endocytosis, causing cytoplasm granulation [150,163]. In fact, the mechanism of non-lytic defensin internalization remains poorly understood [135]. When internalized, defensins can bind intracellular specific targets, inducing signaling cascades. Due to their cationic nature, most defensins are likely to bind nucleic acids which might result in a broad inhibition of DNA synthesis, transcription and/or mRNA translation inside the target cells [167,168]. Such an effect on gene expression could explain the non-morphogenic activity of some defensins and their capacity to interfere with the fungal secondary metabolism, including mycotoxin biosynthesis [127]. One of the most documented defensins for its interaction with intracellular targets is certainly the pea defensin, Psd1. Psd1 has been shown to be translocated to *N. crassa* fungal nucleus and to interact with distinct nuclear proteins including cyclin F and consequently to lead to the disruption of the cell cycle control function in the nuclei [169].

## 5. Exploiting Defensins to Protect Crops from Phytopathogenic Fungi and Mycotoxin Contamination

As illustrated above, several defensins possess efficient and interesting capacities to prevent and/or restrain the growth of phytopathogenic fungi including toxigenic ones and their mechanisms of action have been the subject of numerous investigations. This bioactivity makes defensins promising candidates for consideration in control methods as alternatives to the use of synthetic fungicides. Two application strategies might be explored: the creation of transgenic plants overexpressing antifungal defensins and the formulation of defensin-based plant-care products.

### 5.1. Transgenic Plants Overexpressing Defensin for an Enhanced Resistance to Phytopathogenic Fungi

Gene constructions based on sequences coding for defensins have been expressed in various plant models and/or crops of economic interest. As first reviewed by Montesinos in 2007 [170] and thereafter by Sher Khan et al. [171], these biotechnological developments can provide higher degrees of protection against distinct plant fungal pathogens, either biotrophic, hemi biotrophic or necrotrophic ones. Thus, Gao et al. [81] and Abdallah et al. [172] have reported the increased protection against *F. oxysporum* and *Verticilium dahlia* of potato and tomato plants overexpressing the MsDef1 defensin from *M. sativa*. Similarly, tobacco transformation with MsDef1 led to an improved resistance to *Ralstonia solanacearum* and *A. niger* [173]. DmAMP1 from *D. merckii*, when expressed in papaya, was shown to upscale the resistance to *Phytophthora palmivora* [174] and to reduce symptoms caused by *M. oryzae* and *R. solani* when expressed in rice [175]. The use of RsAFP2 from radish as transgene was demonstrated to enhance tobacco resistance to the pathogen *A. longipes* [68], tomato resistance to *F. oxysporum* [176] and wheat resistance to *R. solani* [177]. Tobacco and potato genetically engineered with NmDef02 from *Nicotiana megalosiphon* were reported to be more tolerant against the oomycete *Phytophthora infestans* [178]. The introduction of the previous NmDef03 transgene was also shown to protect soybean from *Phakopsora pachyrhizi* and *Colletotrichum truncatum* [179]. Lastly, the overexpression of WT1 from *Wasabia japonica* in rice, tomato, potato, egusi melon or tobacco was reported as an efficient strategy to decrease their susceptibility to several phytopathogenic fungi [180,181,182,183]. In several studies, a combination of two defensin genes was used. Thus, the genetic engineering of *A. thaliana* with DmAMP1 and RsAFP1 [184], of rice with DmAMP1 and RsAFP2 [185] and of peanut with NPR1 and Tfgd was successfully experimented. Defensins from non-plant origin were also considered in these biotechnological applications. For example, rice transformation with a transgene related to the fungal defensin AFP from *A. giganteus* was shown to improve plant resistance to the pathogen *M. grisea* [186]. Genetically modified tobacco with genes coding for the arthropod defensins, heliomicin or drosomycin, was reported to exhibit a slight but statistically significant enhanced resistance to the fungal pathogen *Cercospora nicotianae* [187]. Lastly, in a few studies, the inserted DNA fragment contains a defensin gene associated with a non-defensin one. Thus, the co-expression of the RsAFP1 gene and the chitinase Chit42 gene from *Trichoderma atroviride* was demonstrated to enhance canola resistance to sclerotinia stem rot disease [188].

Genetic engineering exploiting the bioactivity of plant defensins could also offer a promising approach for manipulating susceptibility to disease induced by toxigenic fungi and for minimizing mycotoxins in harvests. A small number of defensin transgenes have been explored in order to generate crops that display enhanced resistance or tolerance to Fusarium head blight which is mainly caused by *F. graminearum* or to *Aspergillus* spp. disease. The study of Li et al. [177] described reduced symptoms in wheat lines transformed with RsAFP2 compared to the transgenic control cultivar, cultivated in greenhouse and field trials and artificially inoculated with *F. graminearum*. Similarly, an increased resistance to Fusarium head blight was reported in transgenic wheat lines overexpressing the TAD1 defensin gene [189]. Moreover, the potential of defensin-based engineering strategies to alleviate contamination with mycotoxins was clearly demonstrated in the report of Kaur et al. [190] that indicated significantly reduced amounts of deoxynivalenol in the siliques of Arabidopsis transgenic lines expressing MtDef4.2 that were inoculated with a toxigenic *F. graminearum* strain. MtDef4 and MtDef5 from *M. truncatula* have also been used to boost the resistance of peanut against *A. flavus* and to minimize the contamination of seeds with aflatoxin [151,191].

However, despite the promising results described above, no defensin transgenic plants that confer improved resistance to pathogenic fungi are yet in the market. Indeed, most of the developed countries have set up full and detailed genetically modified organism regulations that require the achievement of a comprehensive risk assessment procedure prior release on the market and this risk assessment is far from being completed with regard to defensin transgenic plants. Moreover, the implementation of field trials also remains highly insufficient to allow concluding on critical issues including reproducibility, stability and environmental effects such as the potential occurrence of side effects affecting the crop productivity. Actually, while the expression of Dm-AMP1 in *Solanum melongena* [192] or MtDef4.2 in wheat [193] was reported as harmless to mycorrhizal fungi, some detrimental effects were also observed in a few defensin transgenic plants. The overexpression of DEF2 was reported to alter the architecture of the tomato plant, to reduce pollen viability as well as seed production [194]. Transgenic *A. thaliana* expressing the plant defensins MsDef1, MtDef2, and RsAFP2, were also negatively affected in their growth, root and root hair development [195]. Lastly, political and ethical concerns related to genetically modified organisms should also not been neglected, representing an additional obstacle that the development of defensin transgenic crops has to overcome before reaching the market.

### 5.2. Developing Defensin-Based Plant Protection Products for the Control of Phytopathogenic Fungi

Given their antifungal efficiency even at low doses, defensins are attractive candidates to replace synthetic fungicides or to reduce their amount by a combinatorial use in plant disease management strategy. The capacity demonstrated by some defensins to inhibit the production of mycotoxins, more precisely of deoxynivalenol [127], is an additional argument in favor of their exploitation in agro-products. Indeed, deoxynivalenol is acknowledged to act as a virulence factor for *F. graminearum* infecting wheat; the fungus used deoxynivalenol production to circumvent the plant’s defense system and invade spikelets [196]. In addition, since deoxynivalenol production is reported as part of the adaptive response of *F. graminearum* to stressful conditions as those induced by exposure to fungicides [197], it is highly recommended that a fungicide solution that also target the production of deoxynivalenol is applied, which will allow avoiding an increased yield of toxins as has been observed with some synthetic fungicide treatments [198]. Moreover, the multifaceted mechanism employed by defensins against fungi is likely to reduce the risk of the emergence of resistant fungal strains through selective pressures [23]. Actually, as exhaustively reviewed by Fisher et al. [199], the emergence of new virulent and fungicide-resistant strains, mainly due to the intensive use of single-target fungicides, has become a critical threat for agriculture of today and tomorrow. Available published data support the fact that AMPs seem to not induce neither antibacterial nor antifungal resistance [200,201]. Furthermore, some fungal defensins were reported to be able to kill antibiotic-resistant bacteria isolates, supporting the promising use of this class of AMPs [202]. Nevertheless, even though unlikely, it cannot be entirely ruled out that phytopathogenic fungi, that are known as remarkable in their ability to adapt in response to selection pressures, could evolve and develop mechanisms to counter the fungicidal action of pesticide, including cell membrane rearrangement, membrane potential and ionic currents change, or peptide degrading enzyme production [41]. Despite increasing evidence supporting the promising use of defensins, the development of defensin-based protection products requires solutions to several hurdles which will be briefly addressed in the following. The first one is the insufficient amount of data supporting the in vivo lack of toxicity of defensins, which hampers a comprehensive assessment of risk and health hazards related to their use as plant protection products and their registration by competent authorities. Indeed, while in vitro cytotoxicity studies converged on the null or reduced the toxic side effects of defensins [203], the body of knowledge that has been developed using animal models remains limited and mainly restricted to defensins of bacterial origin [204].

The second major limitation to the application of defensins for controlling phytopathogenic fungi is the lack of optimized process for their production on a large scale. The yields of defensins from natural sources are generally low and the extraction and purification steps are time-consuming and expensive. Chemical synthesis has for a long time been considered as an economically viable solution but only for short peptides and high-value applications [205]. However, recent advances in peptide synthesis methodologies have paved the way for the successful synthesis of defensins conserving their biological activity and for reducing associated costs. One of the latest successes of defensin chemical synthesis is the production of the PvD1 defensin from the *P. vulgaris* [206]. In recent years, genetic engineering, which is the privileged technology for the production of large amounts of proteins, has been subject of intense investigation for the large-scale production of defensins. Different heterologous expression systems were studied, including *E. coli* [207], yeasts (*Saccharomyces cerevisiae* or *Pichia pastoris*) and insects. Indeed, the use of advanced insect cell-based expression systems was proposed to overcome limitations due to the antimicrobial activity of defensins that could hamper their heterologous production in bacteria and yeasts and to allow the properly synthesis of folded functional peptides which is more challenging using bacteria [208]. In addition, to minimize the lethal effects of the peptide in the host cell, to protect them from proteolytic degradation and improve their solubility, various strategies were elaborated. The most common strategy is based on the use of fusion proteins, associating a defensin and a carrier protein [209,210]. Thus, thioredoxin [211] and small ubiquitin-related modifier [212,213,214,215] have frequently been used as AMPs fusion partner for improving the folding and solubility of the peptide. The promising use of heterologous expression technology to produce defensins and preserve their bioactivity against toxigenic fungal species has been reported by Kant et al. [216] who described the capacity of a recombinant PDC1 corn defensin, expressed in *E. coli* or *P. pastoris*, to inhibit the growth of *F. graminearum*. Interest in the *E. coli* expression system was also recently supported by the study of Al Kashgry et al. [217] which reported the successful production of the MzDef maize defensin and its antifungal activity against *F. verticillioides* and *A. niger*.

Another factor that must be considered for the development of defensin-based plant care products is their stability. As generally small peptides, defensins can be subject to proteolytic degradation by various proteases, resulting in their poor bioavailability and decreased efficacy. However, the intramolecular structure stabilized by disulfide bonds that characterizes defensin makes this class of peptides less proteolytically degradable compared to linear peptides. The occurrence of disulfide bonds has also been reported to confer a high structural stability to defensin at extreme temperatures and pH values [218]. To protect defensins from degradation, improve their solubility and consequently their bioavailability, the use of engineered nano-carriers may be a promising route. Nanoencapsulation systems including micro/nano -suspensions, -emulsions, -particles, -capsules and -hybrids are currently under practice for chemical pesticide application [219], and intensively investigated for medicinal applications of defensins [220].

Last but certainly not least, economic and social acceptance of the use of defensin-based plant fungicides should not be neglected. While integrated pest management practices with less environmental impact including the adoption of biofungicide solutions are convincing an increasing number of farmers [221], the balance between the efficiency and cost of environmentally friendly pesticides can be a barrier for the adoption of these new plant protection solutions. As previously mentioned, efforts should be dedicated to improving the large-scale and low-cost production of defensins and demonstrate their efficiency in field trials. Once these issues are solved, defensin-plant-based solutions will have to be integrated in the framework of policies implemented to change farmer behavior and incentivize the adoption of new practices, which includes advisory services and training, the demonstration of the economic benefits of new and sustainable protection products but also financial support to accompany the transition towards agricultural systems with less use of chemical pesticides [222]. Actually, the adoption and acceptability of defensin-based biopesticides will be impossible without the relevant and wide dissemination of the benefits of their use to stakeholders, which represents a critical step to combat the sometimes negative perception related to new sustainable solutions and avoid their dismissal as a feasible and efficient option for pest management [223]. Defensin-plant-based solutions will also have to meet the requirements for their registration as biofungicides. The term biofungicide mostly refers to fungicides that contain a microorganism as active ingredient, but also involve formulations exploiting the bioactivity of naturally occurring substances. Antifungal peptides with native chemical structure fall within the former definition. The biopesticide registration data portfolio is close to that required for conventional chemical pesticides and includes, among others, information about the mode of action and proof of efficacy, host range testing, toxicological and eco-toxicological evaluations [223]. The guidance of the Organisation for Economic Co-operation and Development (OECD) is that biopesticides should only be authorized if they pose minimal or zero risk. This registration procedure is cumbersome and expensive and can jeopardize the commercialization of a biopesticide such as a defensin-based one if the market seems too small to justify the expenses inherent to its registration. To try solving this issue and boost the development of biopesticides, some countries have modified their legislation so that biological products automatically enter a fast-track review process. This is for instance the case of Canada and the United States, which have implemented a joint review process for biological products whereby a registration dossier receives speedier analysis and once the biopesticide is approved and granted, its commercialization is allowed in both countries simultaneously [224].

## 6. Conclusions

The present review highlights the promising potential of defensins in plant disease treatments to protect crops from phytopathogenic fungi including toxigenic ones. In addition to their efficient antifungal activity and capacity to inhibit the production of mycotoxins, several rationales support the bright future held by this class of natural peptides: defensins exhibit low toxicity to plants and mammals, high stability and solubility, fall within the biopesticide definition and have a possibly low cost of production through microorganism engineering. The development of defensin-based plant protection products could be a new lever to facilitate the transition between current crop production systems based on an intensive use of chemical pesticides towards more sustainable ones. However, despite this outstanding potential, the development of defensin-based biocontrol solutions still faces numerous obstacles. Efforts should be pursued to translate defensin-based in vitro research findings into plant protection products. In addition, the potential offered by defensins in plant disease management is today certainly largely underestimated. Indeed, available knowledge on defensin bioactivity against phytopathogenic fungi is mainly restricted to their antifungal effect and to defensins from plant origin. As previously published [111,113,127], defensins could also exhibit highly promising antimycotoxin efficiency and defensins of invertebrate origin could be an additional source of bioactive peptides. The expansion of peptide libraries and defensin databases, together with the development of bioinformatics and proteomics tools, will certainly contribute to broaden the field of defensin investigation [225].

## Figures and Tables

**Figure 1 jof-08-00229-f001:**
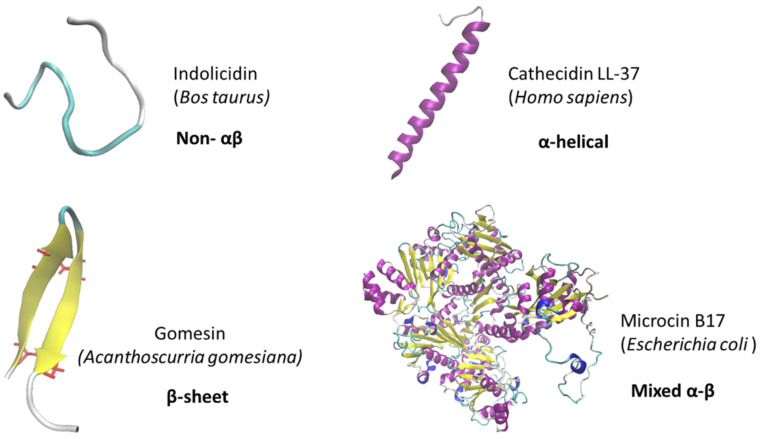
Tridimensional structure of the typical representative of the four groups of AMPs classified according to the presence of α-helix and/or β-sheet secondary elements: non-αβ, α-helical, β-sheet and mixed α-β AMPs.

**Figure 2 jof-08-00229-f002:**
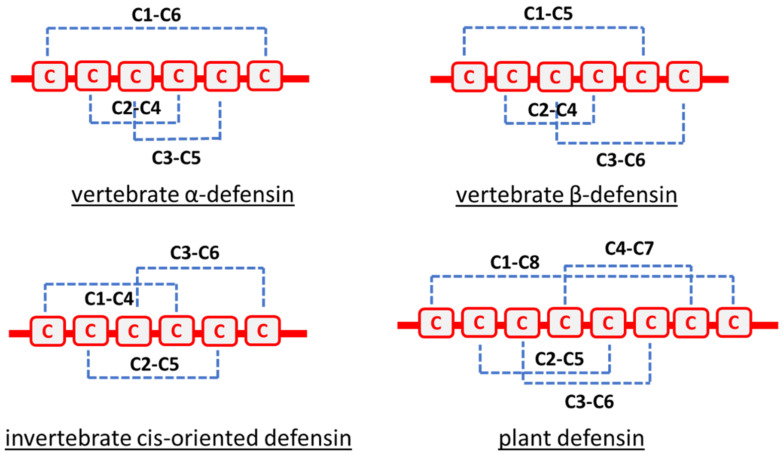
Disulfide bridges’ connectivity pattern characteristic of defensin families: vertebrate α-defensin, vertebrate β-defensin, invertebrate cis-oriented defensin and plant defensin.

**Figure 3 jof-08-00229-f003:**
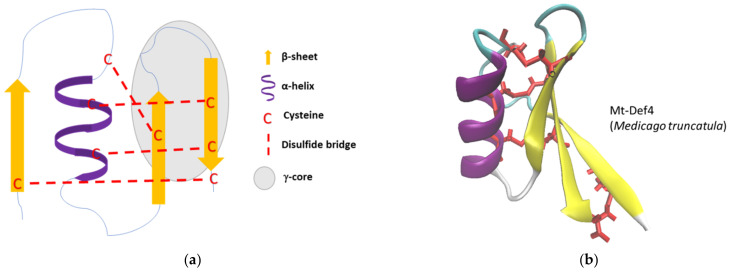
Representation of the cysteine-stabilized alpha-beta (CSαβ) motif present in the structure of plant and of some invertebrate defensins: (**a**) schematic representation of the CSαβ motif present in the plant defensin Mt-Def4; (**b**) schematic representation of the tridimensional structure of the plant defensin Mt-Def4. PDB: 2LR3 from *M. truncatula*. The colors in the figure represent α-helix (purple), β-sheets (yellow), turns (blue) and disulfide bridges (red). The structures were visualized in VMD software version 1.9.3.

**Figure 4 jof-08-00229-f004:**
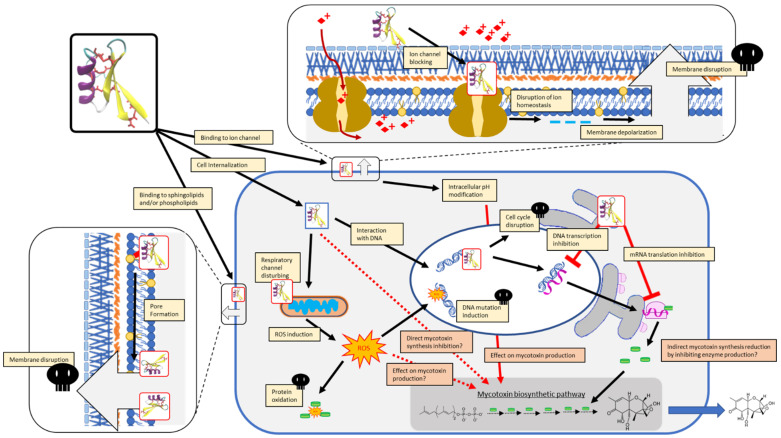
Summary of known and suspected modes of action of defensins displaying antifungal (yellow inserts) and/or antimycotoxin (orange inserts) activity.

**Table 1 jof-08-00229-t001:** List of defensins and DLPs with antifungal effect on phytopathogenic and mycotoxigenic fungi.

Defensin	Amino Acid (AA) Sequence (Accession n°)	AA n°ber	MM (Da)/pI	Organism (Family)	Targeted Species (IC50 and MIC)	Ref.
Plant defensin						
AX1	AICKKPSKFFKGACGRDADCEKACDQENWPGGVCVPFLRCECQRSC (P81493)	44	4895.72/7.14	*Beta vulgaris L* (*Amaran-thaceae*)	*Cercospora beticola* (IC50 = 0.79 µM *)	[62]
AX2	ATCRKPSMYFSGACFSDTNCQKACNREDWPNGKCLVGFKCECQRPC (P82010)	46	5185.01/7.31		*C. beticola* (IC_50_ = 0.39 µM *)	
Dm-AMP1	ELCEKASKTWSGNCGNTGHCDNQCKSWEGAAHGACHVRNGKHMCFCYFNC (P0C8Y4)	46	4997.63/6.87	*Dahlia merckii* (*Asteraceae*)	*B. cinerea* K1147 (IC_50_ = 2.17 µM *); *Cladosporium sphaerospermum* K0791 (IC_50_ = 0.54 µM *); *F. culmorum* K0311 (IC_50_ = (0.18–0.9) µM *); *Leptosphaeria maculans* LM36uea (IC_50_ = 0.27 µM *); *Penicillium digitatum* K0879 (IC_50_ = 0.36 µM *); *Trichoderma viride* K1127 (IC_50_ = 18.1 µM *); *Septoria tritici* K1097 (IC_50_ = 0.18 µM *); *Verticillium alboatrum* K0937 (IC_50_ = 0.72 µM *)	[63]
Dm-AMP2	EVCEKASKTWSGNCGNTGHC	20	2111.33/6.36		*B. cinerea* K1147 (IC_50_ = 1.81 µM *); *C. sphaerospermum* K0791 (IC50 = 0.54 µM *); *F. culmorum* K0311 (IC_50_ = 0.54 µM *); *L. maculans* LM36uea (IC_50_ = 0.18 µM *); *P. digitatum* K0879 (IC_50_ = 0.36 µM *); *T. viride* K1127 (IC_50_ = 18.1 µM *); *S. tritici* K1097 (IC_50_ = 0.18 µM *); *V. albo-atrum* K0937 (IC_50_ = 0.36 µM *)	
AhPDF1.1	QRLCEKPSGTWSGVCGNNGACRNQCIRLEKARHGS	51	5707.65/7.74	*Arabidopsis helleri* (*Brassicaceae*)	*F. oxysporum* (MIC = 0.6 µM)	[64]
At-AFP1	KLCERPSGTWSGVCGNSNACKNQCINLEKARHGSCNYVFPAHKCICYFPC (P30224)	50	5539.44/7.53	*Arabidopsis thaliana* (*Brassicaceae*)	*Alternaria brassicicola* MUCL 20,297 (IC_50_ =1.8 µM *); *B. cinerea* MUCL 30,158 (IC_50_ = 0.7 µM *); *F. culmorum* IMI 180,420 (IC_50_ = 0.54 µM *); *F. oxysporum* f. sp. lycopersici MUCL 909 (IC_50_ = 0.54 µM *); *Pyricularia oryzae* MUCL 30,166 (IC_50_ = 0.05 µM *); *Verticillium dahliae* MUCL 6963 (IC_50_ = 0.27 µM *)	[29]
AtPDF2.3	RTCESKSHRFKGPCVSTHNCANVCHNEGFGGGKCRGFRRRCYCTRHC (Q9ZUL7)	49	5348.15/8.49		*B. cinerea* B05-10 (IC_50_ = 5.8 µM); *B. cinerea* R16 (IC_50_ = 5.8 µM); *F. oxysporum* 5176 (IC_50_ = 4.4 µM); *F. culmorum* MUCL 30,162 (IC_50_ = 1.0 µM); *V. dahliae* MUCL 19,210 (IC_50_ = 4.4 µM); *F. graminearum* PH-1 (IC_50_ = 1.4 µM)	[65]
Hc-AFP1	RYCERSSGTWSGVCGNSGKCSNQCQRLEGAAHGSCNYVFPAHKCICYYPC (G8GZ62)	50	5483.21/7.33	*Heliophila coronopifolia* (*Brassicaceae*)	*B. cinerea* (IC_50_ = 4.56 µM *); *Fusarium solani* (IC_50_ = 4.56 µM *)	[66]
Hc-AFP2	QKLCERPSGTWSGVCGNNNACRNQCINLEKARHGSCNYVFPAHKCICYFPC (G8GZ63)	51	5722.61/7.54		*B. cinerea* (IC_50_ = (1.75–2.62) µM *); *F. solani* (IC_50_ = (1.75–2.62) µM *)	
Hc-AFP3	RYCERSSGTWSGVCGNTDKCSSQCQRLEGAAHGSCNYVFPAHKCICYYPC (G8GZ64)	50	5528.24/7.09		*B. cinerea* (IC_50_ = (3.62–4.52) µM *); *F. solani* (IC_50_ = 4.52 µM *)	
Hc-AFP4	QKLCERPSGTWSGVCGNNGACRNQCIRLERARHGSCNYVFPAHKCICYFPC (G8GZ65)	51	5735.66/7.75		*B. cinerea* (IC_50_ = (2.61–3.49) µM *); *F. solani* (IC_50_ = (0.87–1.74) µM *)	
Rs-AFP1	QKLCERPSGTWSGVCGNNNACKNQCINLEKARHGSCNYVFPAHKCICYFPC (P69241)	51	5694.60/7.53	*R. sativus* (*Brassicaceae*)	*A. brassicicola* MUCL 20,297 (IC_50_ = 2.64 µM *); *B. cinerea* MUCL 30,158 (IC_50_ = 1.41 µM *); *F. culmorum* IMI 180,420 (IC_50_ = 0.88 µM *); *F. oxysporum* f. sp. lycopersici MUCL 909 (IC_50_ = 5.28 µM *); *P. oryzae* MUCL 30,166 (IC_50_ = 0.05 µM *); *V. dahliae* MUCL 6963 (IC_50_ = 0.88 µM *)	[29]
					*Ascochyta pisi* (IC_50_ = 0.88 µM *); *C. beticola* (IC_50_ = 0.35 µM *); *Colletotrichum lindemuthianum* (IC_50_ = 17.61 µM *); *F. oxysporum* f. sp. pisi (IC_50_ = 2.64 µM *); *Mycosphaerella fijiensis* var. fijiensis (IC_50_ = 0.7 µM *); *Nectria haematococca* (IC_50_ = 1.06 µM *); *Phoma betae* (IC_50_ = 0.35 µM *); *Pyrenophora tritici-repentis* (IC_50_ = 0.53 µM *); *P. oryzae* (IC_50_ = 0.05 µM *); *Rhizoctonia solani* (IC_50_ = 17.61 µM *); *Sclerotinia sclerotiorum* (IC_50_ = 3.52 µM *); *Septoria nodorum* (IC_50_ = 3.52 µM *); *Trichoderma hamatum* (IC_50_ = 1.06 µM *); *V. dahliae* (IC_50_ = 0.88 µM *)	[67]
Rs-AFP2	QKLCQRPSGTWSGVCGNNNACKNQCIRLEKARHGSCNYVFPAHKCICYFPC (P30230)	51	5735.70/7.94		*B. cinerea* K1147 (IC_50_ = 1.75 µM *); *C. sphaerospermum* K0791 (IC_50_ = 0.52 µM *); *F. culmorum* K0311 (IC_50_ = 0.26 µM *); *F. culmorum* K0311 (IC_50_ = 0.87 µM *); *L. maculans* LM36uea (IC_50_ = 2.1 µM *); *P. digitatum* K0879 (IC_50_ = 0.26 µM *); *T. viride* K1127 (IC_50_ = 5.25 µM *); *S. tritici* K1097 (IC_50_ = 0.26 µM *); *V. albo-atrum* K0937 (IC_50_ = 2.1 µM *)	[63]
					*A. brassicicola* MUCL 20,297 (IC_50_ = 0.35 µM *); *B. cinerea* MUCL 30,158 (IC_50_ = 0.35 µM *); *F. culmorum* IMI 180,420 (IC_50_ = 0.35 µM *); *F. oxysporum* f. sp. lycopersici MUCL 909 (IC_50_ = 0.35 µM *); *P. oryzae* MUCL 30,166 (IC_50_ = 0.7 µM *); *V. dahliae* MUCL 6963	[29]
					*A. pisi* (IC_50_ = 0.7 µM *); *C. beticola* (IC_50_ = 0.35 µM *); *C. lindemuthianum* (IC_50_ = 0.52 µM *); *F. oxysporum* f. sp. Pisi (IC_50_ = 0.35 µM *); *M. fijiensis* var. fijiensis (IC_50_ = 0.26 µM *); *N. haematococca* (IC_50_ = 0.35 µM *); *P. betae* (IC_50_ = 0.17 µM *); *P. tritici-repentis* (IC_50_ = 0.26 µM *); *R. solani* (IC_50_ = 17.49 µM *); *S. sclerotiorum* (IC_50_ = 17.49 µM *); *S. nodorum* (IC_50_ = 2.62 µM *); *T. hamatum* (IC_50_ = 0.35 µM *); *V. dahliae* (IC_50_ = 0.26 µM *); *Venturia inaequalis* (IC_50_ = 4.37 µM *)	[67]
Defensin-like protein 4	QKLCERSSGTWSGVCGNNNACKNQCINLEGARHGSCNYIFPYHRCICYFPC (O24331)	51	5747.58/7.33		*A. brassicicola* (IC_50_ = 0.87 µM *); *B. cinerea* (IC_50_ = 1.57 µM *); *F. culmorum* (IC_50_ = 1.92 µM *)	[68]
Defensin-like protein 3	KLCERSSGTWSGVCGNNNACKNQCIRLEGAQHGSCNYVFPAHKCICYFPC (O24332)	50	5499.34/7.33		*A. brassicicola* (IC_50_ = 0.36 µM *); *B. cinerea* (IC_50_ = 0.36 µM *); *F. culmorum* (IC_50_ = 0.36 µM *)	
Sa-AFP2	QKLCQRPSGTWSGVCGNNNACRNQCINLEKARHGSCNYVFPAHKCICYFPC (P30232)	51	5721.63/7.74	*S. alba* (*Brassicaceae*)	*A. brassicicola* MUCL 20,297 (IC_50_ = 0.79 µM *); *B. cinerea* MUCL 30,158 (IC_50_ = 0.61 µM *); *F. culmorum* IMI 180,420 (IC_50_ = 0.4 µM *); *F. oxysporum* f. sp. lycopersici MUCL 909 (IC_50_ = 0.4 µM *); *P. oryzae* MUCL 30,166 (IC_50_ = 0.05 µM *); *V. dahliae* MUCL 6963 (IC_50_ = 0.21 µM *)	[29]
WT1	QKLCEKSSGTWSGVCGNNNACKNQCINLEGARHGSCNYIFPYHRCICYFPC (Q9FS38)	51	5719.56/7.33	*Eutrema japonicum* (*Brassicaceae*)	*Magnaporthe grisea* (IC_50_ = 0.87 µM *); *B. cinerea* (IC_50_ = 3.5 µM *)	[69]
Sm-AMP-D1	KICERASGTWKGICIHSNDCNNQCVKWENAGSGSCHYQFPNYMCFCYFDC (C0HL82)	50	5763.55/6.28	*Stellaria media L.* (*Caryophyllaceae*)	*Bipolaris sorokiniana* 6/10 (IC_50_ = 0.5 µM); *F. oxysporum* 16/10 (IC_50_ = 0.35 µM); *F. graminearum* VKM F-1668 (IC_50_ = 0.52 µM); *Fusarium avenaceum* VKM F-2303 (IC_50_ = 0.52 µM); *B. cinerea* SGR-1 (IC_50_ = 1.0 µM); *P. betae* VKM F-2532 (IC_50_ = 0.52 µM); *Pythium debaryanum* VKM F-1505 (IC_50_ = 1.0 µM)	[70]
Sm-AMP-D2	KICERASGTWKGICIHSNDCNNQCVKWENAGSGSCHYQFPNYMCFCYFNC (C0HL83)	50	5762.57/6.77		*B. sorokiniana* 6/10 (IC_50_ = 0.5 µM); *F. oxysporum* 16/10 (IC_50_ = 0.35 µM); *F. graminearum* VKM F-1668 (IC_50_ = 0.52 µM); *F. avenaceum* VKM F-2303 (IC_50_ = 0.52 µM); *B. cinerea* SGR-1 (IC_50_ = 1.0 µM); *P. betae* VKM F-2532 (IC_50_ = 0.52 µM); *P. debaryanum* VKM F-1505 (IC_50_ = 1.0 µM)	
So-D2	GIFSSRKCKTPSKTFKGICTRDSNCDTSCRYEGYPAGDCKGIRRRCMCSKPC	52	5803.79/8.34	*S. oleracea* (*Chenopodiaceae*)	*F. culmorum* (IC_50_ = 0.2 µM); *F. solani* (IC_50_ = 11 µM); *Colletotrichum lagenarium* (IC_50_ = 11 µM); *Bipolaris maydis* (IC_50_ = 6 µM)	[71]
AB2	RTCENLANTYRGPCITTGSCDDHCKNKEHLRSGRCRDDFRCW	47	5469.18/7.33	*Adzuckia angularia* (*Fabaceae*)	*B. cinerea* (IC_50_ = 3.5 µM)	[72]
Beta-astratide bM1	CEKPSKFFSGPCIGSSGKTQCAYLCRRGEGLQDGNCKGLKCVCAC	45	4734.58/7.52	*Astragalus membranaceus* (*Fabaceae*)	*F. oxysporum* CICC 2532 (IC_50_ = 4.92 µM *); *Alternaria alternata* CICC 2465 (IC_50_ = 4.75 µM *); *R. solani* CICC 40,259 (IC_50_ = 27.52 µM *); *Curvularia lunata* CICC 40,301 (IC_50_ = 0.57 µM *)	[73]
Coccinin	KQTENLADTY (P84785)	10	1182.25/4.19	*Phaseolus coccineus* cv. ‘Major’ (*Fabaceae*)	*F. oxysporum* (MIC = 81 µM); *B. cinerea* (MIC = 109 µM); *R. solani* (MIC = 134 µM); *Mycosphaerella arachidicola* (MIC = 75 µM)	[74]
Phaseococcin	KTCENLADTYKGPPPFFTTG	20	2187.46/6.03	*P. coccineus* cv. ‘Minor’ (*Fabaceae*)	*F. oxysporum* (MIC = 89 µM); *B. cinerea* (MIC = 102 µM); *R. solani* (MIC = 140 µM); *M. arachidicola* (MIC = 70 µM)	[75]
Ct-AMP1	NLCERASLTWTGNCGNTGHCDTQCRNWESAKHGACHKRGNWKCFCYFNC (Q7M1F2)	49	5613.32/7.33	*Clitoria ternatea* (*Fabaceae*)	*B. cinerea* K1147 (IC_50_ = 3.56 µM *); *C. sphaerospermum* K0791 (IC_50_ = 1.07 µM *); *F. culmorum* K0311 (PDB medium) (IC_50_ = 1.78 µM *); *F. culmorum* K0311 (SMF medium) (IC_50_ = 0.11 µM *); *L. maculans* LM36uea (IC_50_ = =1.07 µM *); *P. digitatum* K0879 (IC_50_ = 3.56 µM *); *T. viride* K1127 (IC_50_ = 17.81 µM *); *S. tritici* K1097 (IC_50_ = 0.36 µM *); *V. albo-atrum* K0937 (IC_50_ = 0.36 µM *)	[63]
Gymnin	KTCENLADDY (P84200)	10	1171.25/3.8	*Gymnocladus chinensis* (*Fabaceae*)	*F. oxysporum* (IC_50_ = 2 µM); *Cercospora arachidicola* (IC_50_ = 10 µM)	[76]
Lc-def	KTCENLSDSFKGPCIPDGNCNKHCKEKEHLLSGRCRDDFRCWCTRNC (B3F051)	47	5449.23/7.08	*Lens culinaris* (*Fabaceae*)	*Aspergillus niger* VKM F-2259 (IC_50_ = 18.5 µM); *Aspergillus versicolor* VKM F-1114 (IC_50_ = 18.5 µM); *B. cinerea* VKM F-3700 (IC_50_ = 9.25 µM); *F. culmorum* VKM F-844 (IC_50_ = (18.5–37.0) µM)	[77]
Limenin	KTCENLADTYKGPCFTTGGCDDHCKNKEHLLSGRCRDDFRCWCTRNC	47	5403.12/6.77	*Phaseolus limensis* (*Fabaceae*)	*B. cinerea* (MIC = 2.9 µM); *F. oxysporum* (MIC = 2.1 µM); *M. arachidicola* (MIC = 0.34 µM)	[78]
Limyin	KTCENLATYYRGPCF	15	1766.03/7.51		*F. solani* (IC_50_ = 8.6 µM)	[79]
Ms-Def1 (alfAFP)	RTCENLADKYRGPCFSGCDTHCTTKENAVSGRCRDDFRCWCTKRC (Q4G3V1)	45	5194.90/7.32	*Medicago sativa* (*Fabaceae*)	*F. graminearum* (IC_50_ = (1.2–2.3) µM *)	[80]
					F. graminearum PH-1 (IC_50_ = (2–4) µM *); *F. graminearum* PH-1 (MIC > 6 µM)	[56]
					*V. dahliae* (MIC = 1 µM *)	[81]
Mt-Def2	KTCENLADKYRGPCFSGCDTHCTTKENAVSGRCRDDFRCWCTKRC (Q5YLG8)	45	5166.89/7.32	*M. Truncatula* (*Fabaceae*)	*F. graminearum* PH-1 (IC_50_ = (0.75–1) µM)	[56]
					*F. oxysporum* f. sp. medicaginis 7F-3 (IC_50_ = 0.7 µM); *F. oxysporum* f. sp. medicaginis 31F-3 (IC_50_ = 1.9 µM); *Phoma medicaginis* STC (IC_50_ = 0.3 µM); *P. medicaginis* WS-2 (IC_50_ = 2.6 µM); *Clavibacter insidiosus* (IC_50_ = 0.1 µM)	[35]
Mt-Def4	RTCESQSHKFKGPCASDHNCASVCQTERFSGGRCRGFRRRCFCTTHC (G7L736)	47	5343.08/7.97		*F. graminearum* PH-1 (IC_50_ = (0.75–1) µM)	[56]
					*F. oxysporum* f. sp. medicaginis 7F-3 (IC_50_ = 0.7 µM); *F. oxysporum* f. sp. medicaginis 31F-3 (IC_50_ = 1.9 µM); *P. medicaginis* STC (IC_50_ = 0.3 µM); *P. medicaginis* WS-2 (IC_50_ = 2.6 µM)	[35]
PsD1	KTCEHLADTYRGVCFTNASCDDHCKNKAHLISGTCHNWKCFCTQNC (P81929)	46	5208.93/6.81	*Pisum sativum* (*Fabaceae*)	*A. niger* EK0197 (IC_50_ = 2.3 µM *); *A. versicolor* 40028LMR/INCQS (IC_50_ = 1 µM *); *Fusarium moniliforme* 2414UFPe (IC_50_ = 4.2 µM *); *F. oxysporum* 2665UFPe (IC_50_ = 19.2 µM *); *F. solani* 2389UFPe (IC_50_ = 2.3 µM *)	[82]
PsD2	KTCENLSGTFKGPCIPDGNCNKHCRNNEHLLSGRCRDDFRCWCTNRC (P81930)	47	5404.15/7.33		*A. niger* EK0197 (IC_50_ = 1.9 µM *); *A. versicolor* 40028LMR/INCQS (IC_50_ = 0.06 µM *); *F. moniliforme* 2414UFPe (IC_50_ = 1.85 µM *); *F. oxysporum* 2665UFPe (IC_50_ = 18.5 µM *); F. solani 2389UFPe (IC_50_ = 1.57 µM *)	
PvD1_PTA2c	KTCENLADTYKGPCFTTGSCDDHCKNKEHLRSGRCRDDFRCWCTKNC (F8QXP9)	47	5448.16/7.08	*Phaseolus vulgaris* (*Fabaceae*)	*F. solani* (IC_50_ = 18.35 µM *); *Fusarium laterithium* (IC_50_ = 18.35 µM *); *R. solani* (IC_50_ = 18.35 µM *); *F. oxysporum* (IC_50_ = 18.35 µM *)	[83]
					*B. cinerea* (IC_50_ = 1 µM)	[72]
P. vulagris white cloud defensin	KTCENLADTFRGPCFATSNCDDHCKNKEHLLSGRCRDDFRCWCTRNC	47	5472.18/6.77	*P. vulgaris* cv. “white cloud bean” (*Fabaceae*)	*B. cinerea* (MIC = 2.8 µM); *F. oxysporum* (MIC = 2.3 µM); *M. arachidicola* (MIC = 0.72 µM)	[84]
Sesquin	KTCENLADTY (P84868)	10	1157.27/4.19	*Vigna unguiculate* (*Fabaceae*)	*B. cinerea* (IC_50_ = 2.5 µM); *F. oxysporum* (IC_50_ = 1.4 µM); *M. arachidicola* (IC_50_ = 0.15 µM)	[85]
SPE10	KTCENLADTFRGPCFTDGSCDDHCKNKEHLIKGRCRDDFRCWCTRNC (Q6B519)	47	5500.24/6.77	*Pachyrhizus erosus* (*Fabaceae*)	*Aspergillus flavus* (IC_50_ = 5.45 µM *); *A. niger* (IC_50_ = 8.18 µM *); *B. maydis* (IC_50_ = 2.73 µM *); *B. cinerea* (IC_50_ = 18.18 µM *); *Colletotrichum gloeosporides* (IC_50_ = 18.18 µM *); *F. oxysporum* f.sp. lycopersic (IC_50_ = 18.18 µM *); *F. oxysporum* f.sp. vasinfectum (IC_50_ = 18.18 µM *); *Penicillium* spp. (IC_50_ = 18.18 µM *); *Rhizopus stolonifer* (IC_50_ = 18.18 µM *); *V. dahliae* (IC_50_ = 18.18 µM *)	[86]
TvD1	KTCENLADTYRGPCFTTGSCDDHCKNKEHLLSGRCRDDFRCWCTKRC (Q2KM12)	47	5475.23/7.09	*Tephrosia villosa* (*Fabaceae*)	*Nothopassalora personata* (MIC = 2.05 µM *); *F. oxysporum* (MIC = 5.12 µM *); *Fusarium verticillioides* (MIC = 5.12 µM *); *B. cinerea* (MIC = 5.12 µM *); *Curvularia sp* (MIC = 5.12 µM *); *R. solani* (MIC = 7.78 µM *)	[87]
VaD1	KTCMTKKEGWGRCLIDTTCAHSCRKQGYKGGNCKGMRRTCYCLLDC (A0A0S3QXX7)	46	5209.23/8.12	*Vigna angularis* (*Fabaceae*)	*F. oxysporum* (IC_50_ = 5.76 µM *); *F. oxysporum* f. sp. pisi (IC_50_ = 10.21 µM *)	[88]
VrD1	RTCMIKKEGWGKCLIDTTCAHSCKNRGYIGGNCKGMTRTCYCLVNC (Q6T418)	46	5122.15/7.92	*Vigna radiata* (*Fabaceae*)	*F. oxysporum* (IC_50_ = 1.1 µM *); *F. oxysporum* CCRC 35,270 (IC_50_ = 3.4 µM *); *F. oxysporum* f. sp. Pisi (IC_50_ = 2.4 µM *); *P. oryzae* (IC_50_ = 4 µM *); *R. solani* (IRTCENLADKYRGPCFSGCDTHCTTKENAVSGRCRDDFRCWCTKRCC_50_ = 17.7 µM *)	[89]
PgD1	RTCKTPSGKFKGVCASSNNCKNVCQTEGFPSGSCDFHVANRKCYCSKPCP (Q6RSS6)	50	5377.21/7.91	*Picea glauca* (*Pinaceae*)	*Nectria galligena* (MIC = 2.6 µM *); *F. oxysporum* (MIC = 2.6 µM *)	[51]
PgD5	RMCESQSHKFKGYCASSSNCKVVCQTEKFLTGSCRDTHFGNRRCFCEKPC	50	5729.62/7.72		*F. oxysporum* (MIC = 1.92 µM *); *V. dahliae* (MIC = 0.35 µM *); *B. cinerea* (MIC = 0.7 µM *)	
PsDef1	RMCKTPSGKFKGYCVNNTNCKNVCRTEGFPTGSCDFHVAGRKCYCYKPCP (A4L7R7)	50	5601.58/8.12	*Pinus sylvestris* (*Pinaceae*)	*F. solani* UKM F-50639 (IC_50_ = 0.16 µM *); *F. oxysporum* UKM F-52897 (IC_50_ = 0.52 µM *); *B. cinerea* UKM F-16753 (IC_50_ = 0.07 µM *)	[90]
Ec-AMP-D1	RECQSQSHRYKGACVHDTNCASVCQTEGFSGGKCVGFRGRCFCTKAC (P86518)	47	5107.82/7.54	*Echinochloa crusgalli* (*Poaceae*)	*F. graminearum* (IC_50_ = 2.94 µM *); *F. verticillioides* (IC_50_ = 1.66 µM *); *Diplodia maydis* (IC_50_ = 2.45 µM *); *F. oxysporum* (IC_50_ = 19.97 µM *)	[91]
Ec-AMP-D2	RECQSQSHRYKGACVHDTNCASVCQTEGFSGGKCVGFRGRCFCTKHC (P86519)	47	5173.89/7.54		*F. oxysporum* (IC_50_ = 19.71 µM *)	
Pp-AMP1	KSCCRSTQARNIYNAPRFAGGSRPLCALGSGCKIVDDKKTPPND	44	4697.39/8.61	*Phyllostachys pubescens* (*Poaceae*)	*F. oxysporum* IFO 6384 (IC_50_ = 0.43 µM *)	[92]
Pp-AMP2	KSCCRSTTARTARVPCAKKSNIYNGCRVPGGCKIQEAKKCEPPYD	45	4919.76/8.52		*F. oxysporum* IFO 6384 (IC_50_ = 0.41 µM *)	
Sd1	RYCLSQSHRFKGLCMSSSNCANVCQTENFPGGECKADGATRKCFCKKIC (B2CNV2)	49	5412.32/7.72	*Saccharum officinarum* (*Poaceae*)	*A. niger* (IC_50_ = 2.0 µM); *F. solani* (IC_50_ = 1.0 µM)	[93]
Sd3	RHRHCFSQSHKFVGACLRESNCENVCKTEGFPSGECKWHGIVSKCHCKRIC	51	5864.82/7.73		*A. niger* (IC_50_ = 1.0 µM); *F. solani* (IC_50_ > 20 µM)	
Sd5	HTPTPTPICKSRSHEYKGRCIQDMDCNAACVKESESYTGGFCNGRPPFKQCFCTKPCKRERAAATLRWPGL (A0A1B3B2K6)	71	7967.21/7.91		*A. niger* (IC_50_ > 20 µM); *F. solani* (IC_50_ = 10 µM)	
SI alpha-1	RVCMGKSQHHSFPCISDRLCSNECVKEEGGWTAGYCHLRYCRCQKAC (P21923)	47	5382.26/7.33	*Sorghum bicolor* (*Poaceae*)	*B. cinerea* K1147 (IC_50_ = 18.58 µM *); *C. sphaerospermum* K0791 (IC_50_ = 14.86 µM *); *F. culmorum* K0311 (IC_50_ = 37.16 µM *); *P. digitatum* K0879 (IC_50_ = 37.16 µM *); *T. viride* K1127 (IC_50_ = 9.29 µM *)	[63]
Tk-AMP-D1	RTCQSQSHKFKGACFSDTNCDSVCRTENFPRGQCNQHHVERKCYCERDC (P84963)	49	5744.40/7.1	*Triticum kiharae* (*Poaceae*)	*F. graminearum* (IC_50_ = 5.22 µM *); *F. verticillioides* (IC_50_ = 5.22 µM *)	[91]
ZmD32	RTCQSQSHRFRGPCLRRSNCANVCRTEGFPGGRCRGFRRRCFCTTHC (A0A317Y7J2)	47	5466.33/10.85	*Zea mays* (*Poaceae*)	*F. graminearum* PH-1 (IC_50_ = 1 µM)	[94]
ZmESR6	KLCSTTMDLLICGGAIPGAVNQACDDTCRNKGYTGGGFCNMKIQRCVCRKPC (D1MAH4)	52	5516.57/7.52		*F. oxysporum* f.sp. Conglutinans (IC_50_ = 3 µM); *F. oxysporum* f.sp.lycopersici (IC_50_ = 3 µM); *Plectosphaerella cucumerina* (IC_50_ = 2 µM)	[95]
Fa-AMP1	AQCGAQGGGATCPGGLCCSQWGWCGSTPKYCGAGCQSNCK (P0DKH7)	40	3887.42/7.07	*Fagopyrum esculentum* (*Polygonaceae*)	*F. oxysporum* IFO 6384 (IC_50_ = 4.89 µM *)	[96]
Fa-AMP2	AQCGAQGGGATCPGGLCCSQWGWCGSTPKYCGAGCQSNCR (P0DKH8)	40	3915.44/7.07		*F. oxysporum* IFO 6384 (IC_50_ = 7.41 µM *)	
Ns-D1	KFCEKPSGTWSGVCGNSGACKDQCIRLEGAKHGSCNYKPPAHRCICYYEC (P86972)	50	5487.32/7.32	*Nigella sativa* (*Ranunculaceae*)	*A. niger* VKM F-33 (IC_50_ = 0.64 µM *); *B. sorokiniana* VKM F-1446 (IC_50_ = 0.55 µM *); *F. oxysporum* (IC_50_ = 1.73 µM *); *F. graminearum* VKM F-1668 (IC_50_ = 1.26 µM *; *F. culmorum* VKM F-2303 (IC_50_ = 1.26 µM *); *B. cinerea* (IC_50_ = 4.99 µM *)	[97]
Ns-D2	KFCEKPSGTWSGVCGNSGACKDQCIRLEGAKHGSCNYKLPAHRCICYYEC (P86973)	50	5503.36/7.32		*A. niger* VKM F-33 (IC_50_ = 0.64 µM *); *B. sorokiniana* VKM F-1446 (IC_50_ = 0.33 µM *); *F. oxysporum* (IC_50_ = 0.96 µM *); *F. graminearum* VKM F-1668 (IC_50_ = 1.25 µM *); *F. culmorum* VKM F-2303 (IC_50_ = 1.25 µM *); *B. cinerea* (IC_50_ = 2.49 µM *)	
Ah-AMP1	LCNERPSQTWSGNCGNTAHCDKQCQDWEKASHGACHKRENHWKCFCYFNC (Q7M1F3)	50	5863.53/6.82	*Aesculus hippocastanum* (*Sapindaceae*)	*B. cinerea* K1147 (IC_50_ = 4.26 µM *); *C. sphaerospermum* K0791 (IC_50_ = 0.85 µM *); *F. culmorum* K0311 (IC_50_ = 2.05 µM *); *F. culmorum* K0311 (IC_50_ = 0.12 µM *); *L. maculans* LM36uea (IC_50_ = 0.09 µM *); *P. digitatum* K0879 (IC_50_ = 1.02 µM *); *T. viride* K1127 (IC_50_ = 17.05 µM *); *S. tritici* K1097 (IC_50_ = 0.85 µM *); *V. albo-atrum* K0937 (IC_50_ = 1.02 µM *)	[63]
Hs-AFP1	DGVKLCDVPSGTWSGHCGSSSKCSQQCKDREHFAYGGACHYQFPSVKCFCKRQC (P0C8Y5)	54	5948.76/7.32	*Heuchera sanguinea* (*Saxifragaceae*)	*B. cinerea* K1147 (IC_50_ = 1 µM *); *C. sphaerospermum* K0791 (IC_50_ = 0.2 µM *); *F. culmorum* K0311 (IC_50_ = 0.2 µM *); *L. maculans* LM36uea (IC_50_ = 4.2 µM *); *P. digitatum* K0879 (IC_50_ = 0.2 µM *); *T. viride* K1127 (IC_50_ = 2.5 µM *); *S. tritici* K1097 (IC_50_ = 0.1 µM *); *V. albo-atrum* K0937 (IC_50_ = 1 µM *)	
NaD1	RECKTESNTFPGICITKPPCRKACISEKFTDGHCSKILRRCLCTKPC (Q8GTM0)	47	5304.37/7.91	*Nicotiana alata* (*Solanaceae*)	*A. niger* 5181 (IC_50_ = 2.1 ± 0.76 µM); *A. flavus* 5310 (IC_50_ > 10 µM); *F. oxysporum* f.sp. Vasinfectum (IC_50_ = 1.5 ± 0.25 µM); *F. graminearum* (IC_50_ = 0.4 ± 0.3 µM); *Colletotrichum graminicola* (IC_50_ = 4.4 ± 0.1 µM); *Aspergillus parasiticus* 4467 (IC_50_ = 4.5 ± 0.27 µM)	[98]
					*V. dahliae* (IC_50_ = 0.75 µM); *Thielaviopsis basicola* (IC_50_ = 1 µM); *Aspergillus nidulans* (IC_50_ = 0.8 µM); *Puccinia coronata* f.sp. Avenae (IC_50_ = 2.5 µM); *Puccinia sorghi* (IC_50_ = 2 µM)	[99]
NaD2	RTCESQSHRFKGPCARDSNCATVCLTEGFSGGDCRGFRRRCFCTRPC (A0A1B2YLI5)	47	5264.02/7.76		*F. oxysporum* f.sp. Vasinfectum (IC_50_ = 8.3 µM); *F. graminearum* (IC_50_ = 2 µM); *V. dahliae* (IC_50_ > 10 µM); *T. basicola* (IC_50_ = 7 µM); *A. nidulans* (IC_50_ = 5 µM); *P. coronata* f.sp. Avenae (IC_50_ = 4 µM); *P. sorghi* (IC_50_ = 5 µM)	
PhD1	ATCKAECPTWDSVCINKKPCVACCKKAKFSDGHCSKILRRCLCTKEC (Q8H6Q1)	47	5211.33/7.67	*Petunia hybrida* (*Solanaceae*)	*F. oxysporum* (MIC = (0–0.38) µM *); *B. cinerea* (MIC = (0.38–1.92) µM *)	[100]
PhD2	GTCKAECPTWEGICINKAPCVKCCKAQPEKFTDGHCSKILRRCLCTKPC (Q8H6Q0)	49	5403.55/7.52		*F. oxysporum* (MIC = (0.38–1.92) µM *); *B. cinerea* (MIC = (0.38–1.92) µM *)	
Vv-AMP1	RTCESQSHRFKGTCVRQSNCAAVCQTEGFHGGNCRGFRRRCFCTKHC (D7TAI4)	47	5355.13/8.24	*Vitis vinifera* (*Vitaceae*)	*F. oxysporum* ATCC 10,913 (IC_50_ = 1.12 µM *); *V. dahliae* ATCC 96,522 (IC_50_ = 0.34 µM *); *F. solani* (IC_50_ = 1.79 µM *); *B. cinerea* (IC_50_ = 2.43 µM *)	[101]
Invertebrate defensin						
AgDef1	ATCDLASGFGVGSSLCAAHCIARRYRGGYCNSKAVCVCRN (B2FZB7)	40	4141.80/7.82	*Anopheles gambiae* (*Insecta*)	*F. culmorum* (MIC = (3–6) µM); *F. oxysporum* (MIC = (1.5–3) µM)	[30]
Defensin ARD1	DKLIGSCVWGAVNYTSNCNAECKRRGYKGGHCGSFANVNCWCET (P84156)	44	4803.43/7.24	*Archaeoprepona demophon* (*Insecta*)	*Aspergillus fumigatus* GASP 4707 (MIC = 2.6 µM *)	[102]
DEFC	ATCDLLSGFGVGDSACAAHCIARRNRGGYCNAKKVCVCRN (P81603)	40	4161.84/7.81	*Aedes aegypti* (*Insecta*)	*F. culmorum* (MIC = (50–100) µM)	[30]
Drosomycin	DCLSGRYKGPCAVWDNETCRRVCKEEGRSSGHCSPSLKCWCEGC (P41964)	44	4897.59/6.75	*Drosophila melanogaster* (*Insecta*)	*B. cinerea* MUCL 30,158 (IC_50_ = 1.2 µM); *F. culmorum* IMI 180,420 (IC_50_ = 1.0 µM); *A. brassicicola* MUCL 20,297 (IC_50_ = 0.9 µM); *Alternaria longipes* CBS 62,083 (IC_50_ = 1.4 µM); *N. haematococca* CollectionVanEtten160-2-2 (IC_50_ = 1.8 µM); *F. oxysporum* MUCL 909 (IC_50_ = 4.2 µM); *A. pisi* MUCL 30,164 (IC_50_ = 3.2 µM)	[103]
Gm defensin-like peptide	DKLIGSCVWGATNYTSDCNAECKRRGYKGGHCGSFWNVNCWCEE (P85215)	44	4949.53/6.21	*Galleria mellonella* (*Insecta*)	*A. niger* (MIC = (1.4–2.9) µM); *Trichoderma harzianum* (MIC = (1.4–2.9) µM)	[104]
Galleria defensin	DTLIGSCVWGATNYTSDCNAECKRRGYKGGHCGSFLNVNCWCE (P85213)	43	4720.29/6.2		*F. oxysporum* (MIC = (8.5–16.9) µM); *A. niger* (MIC = (1.1–2.1) µM); *T. harzianum* (MIC = (2.1–4.2) µM)	
Heliomicin	DKLIGSCVWGAVNYTSDCNGECKRRGYKGGHCGSFANVNCWCET (D3G9G5)	44	4790.39/6.87	*Heliothis virescens* (*Insecta*)	*F. culmorum* IMI 180,420 (MIC = (0.2–0.4) µM); *F. oxysporum* MUCL 909 (MIC = (1.5–3.0) µM); *N. haematococca* 160.2.2 (MIC = (0.4–0.8) µM); *A. fumigatus* (MIC = (6–12) µM); *T. viride* MUCL 19,724 (MIC = (1.5–3) µM)	[105]
PduDef	ATCDLLSAFGVGHAACAAHCIGHGYRGGYCNSKAVCTCRR (P83404)	40	4101.74/7.55	*Phlebotomus duboscqi* (*Insecta*)	*A. fumigatus* (MIC = 12.5–25 µM); *F. culmorum* (MIC = 1.56–3.12 µM); *F. oxysporum* (MIC = 3.12–6.25 µM); *T. viride* (MIC = 3.12–6.25 µM)	[30]
Phormicin	ATCDLLSGTGINHSACAAHCLLRGNRGGYCNGKGVCVCRN (P10891)	40	4066.69/7.55	*Protophormia terraenovae* (*Insecta*)	*F. culmorum* IMI 180,420 (MIC = 3 µM); *F. oxysporum* MUCL 909 (MIC = 6 µM)	[105]
					*F. culmorum* (MIC = (1.5–3.0) µM); *F. oxysporum* (MIC = (3–6) µM); *N. haematococca* (MIC = (0.8–1.5) µM); *T. viride* (MIC = (6–12) µM)	[106]
PxDef	RIPCQYEDATEDTICQQHCLPKGYSYGICVSYRCSCV	37	4233.84/5.27	*Plutella xylostella* (*Insecta*)	*B. cinerea* (MIC = 15.0 µM); *Penicillium crustosum* (MIC = 13.0 µM); *Colletotrichum gloeosporioides* Penz. (MIC = 17.3 µM); *Colletotrichum orbiculare* (MIC = 12.5 µM); *F. oxysporum* (MIC = 8.0 µM)	[107]
Royalisin	VTCDLLSFKGQVNDSACAANCLSLGKAGGHCEKGVCICRKTSFKDLWDKRF (P17722)	51	5525.45/7.5	*Apis mellifera* (*Insecta*)	*B. cinerea* (MIC = 4.9 µM *)	[108]
Termicin	ACNFQSCWATCQAQHSIYFRRAFCDRSQCKCVFVRG (P82321)	36	4221.89/7.82	*Pseudacanthotermes springer* (*Insecta*)	*F. culmorum* (MIC = (0.2–0.4) µM); *F. oxysporum* (MIC = (0.8–1.5) µM); *N. haematococca* (MIC = (0.05–0.1) µM); Trichoderma viridae (MIC = (6–12) µM)	[106]
Cg-Def	GFGCPGNQLKCNNHCKSISCRAGYCDAATLWLRCTCTDCNGKK (Q4GWV4)	43	4642.40/7.53	*Crassostrea gigas* (*Bivalvia*)	*B. cinerea* (MIC > 20 µM); *P. crustosum* (MIC > 20 µM); *F. oxysporum* (MIC = 9 µM)	[109]
MGD-1	GFGCPNNYQCHRHCKSIPGRCGGYCGGWHRLRCTCYRC (P80571)	38	4351.07/7.99	*Mytilus galloprovincialis* (*Bivalvia*)	*F. oxysporum* (MIC = 5 µM)	[110]
DefMT3	GYYCPFRQDKCHRHCRSFGRKAGYCGNFLKRTCICVKK (A0A089VRA3)	38	4531.39/9.09	*Ixodes ricinus* (*Arachnida*)	*F. culmorum* (IC_50_ = 4 µM); *F. graminearum* 8/1 (IC_50_ = 4 µM)	[36]
DefMT5	GFFCPYNGYCDRHCRKKLRRRGGYCGGRWKLTCICIMN	38	4533.43/9.11		*F. culmorum* (IC_50_ = 4 µM); *F. graminearum* 8/1 (IC_50_ = 4 µM)	
DefMT6	GFGCPLNQGACHNHCRSIKRRGGYCSGIIKQTCTCYRK	38	4217.95/8.76		*F. culmorum* (IC_50_ = 12 µM); *F. graminearum* 8/1 (IC_50_ = 2 µM)	
Holosin 2	GFGCPLNQRACHRHCRSIGRRGGFCAGLIKQTCTCYRK (A0A5C1Z8V5)	38	4256.05/9.39	*Ixodes holocyclus* (*Arachnida*)	*F. graminearum* PH-1 (MIC = 5 µM)	[111]
Holosin 3	GFGCPNEWRCNAHCKRNRFRGGYCDSWFRRRCHCYG (A0A5C1ZAY3)	36	4400.01/8.59		*F. graminearum* PH-1 (MIC = 5 µM)	
Juruin	FTCAISCDIKVNGKPCKGSGEKKCSGGWSCKFNVCVKV (B3EWQ0)	38	4012.80/7.99	*Avicularia juruensis* (*Arachnida*)	*A. niger* (MIC= (5–10) µM)	[112]
Scapularisin-3	AFGCPFDQGTCHSHCRSIRRRGERCSGFAKRTCTCYQK (B7Q4Z2)	38	4355.00/8.49	*Ixodes scapularis* (*Arachnida*)	*F. culmorum* (IC_50_ = 0.5 µM); *F. graminearum* 8/1 (IC_50_ = 1 µM)	[113]
Scapularisin-6	GFGCPFDQGACHRHCQSIGRRGGYCAGFIKQTCTCYHN (Q5Q979)	38	4180.76/7.55		*F. culmorum* (IC_50_ = 1 µM); *F. graminearum* 8/1 (IC_50_ = 2 µM)	
Vertebrate defensin						
Hepcidin-1 (Hepcidin-6)	CRFCCRCCPRMRGCGLCCRF	20	2374.04/7.77	*Acanthopagrus schlegelii* (*Sparidae*)	*A. niger* CGMCC 3.316 (MIC = 20–40 µM); *F. graminearum* CGMCC 3.3490 (MIC = 20–40 µM); *F. solani* CGMCC 3.5840 (MIC = 20–40 µM)	[114]
Hepcidin-2	SPAGCRFCCGCCPNMRGCGVCCRF (Q68M56)	24	2531.12/7.33		*A. niger* CGMCC 3.316 (MIC= 40–60 µM); *F. graminearum* CGMCC 3.3490 (MIC > 60 µM); *F. solani* CGMCC 3.5840 (MIC > 60 µM)	
Hepcidin	GCRFCCNCCPNMSGCGVCCRF (P82951)	21	2263.79/7.08		*A. niger* (MIC = 44 µM)	[115]
Crotamine	YKQCHKKGGHCFPKEKICLPPSSDFGKMDCRWRWKCCKKGSG (Q9PWF3)	42	4889.85/8.58	*Crotalus durissus terrificus* (*Viperidae*)	*A. fumigatus* IOC 4526 (MIC >25.5 µM *)	[116]
Spheniscin-2	SFGLCRLRRGFCARGRCRFPSIPIGRCSRFVQCCRRVW (P83430)	38	4507.47/11.47	*Aptenodytes patagonicus* (*Spheniscidae*)	*A. fumigatus* (MIC= (3–6) µM)	[117]
Human drosomycin-like defensin	CLAGRLDKQCTCRRSQPSRRSGHEVGRPSPHCGPSRQCGCHMD	43	4751.43/8.15	*Homo sapiens* (*Hominidae*)	*A. fumigatus* ATCC MYA1163 (MIC = 6.25 µM); *A. nidulans* AZN 2867 (MIC = 6.25 µM); *Aspergillus ustus* (MIC = 12.5 µM); *F. solani* AZN 6836 (MIC = 25 µM); *F. oxysporum* (MIC = 6.25 µM)	[118]
Fungus defensin						
AFP	ATYNGKCYKKDNICKYKAQSGKTAICKCYVKKCPRDGAKCEFDSYKGKCYC (P17737)	51	5805.86/8.34	*Aspergillus giganteus* (*Trichocomaceae*)	*Fusarium sporotrichioides* IfGB 39/1601 (MIC = 0.02 µM *); *F. moniliforme* IfGB 39/1402 (MIC = 0.02 µM *); *A. niger* ATCC 9029 (MIC = 0.17 µM *); *A. niger* NRRL 372 (MIC = 0.17 µM *); *A. niger* IfGB 15/1803 (MIC = 0.17 µM *); *Fusarium equiseti* IfGB 39/0701 (MIC = 0.17 µM *); *Fusarium lactis* IfGB 39/0701 (MIC = 0.17 µM *); *F. oxysporum* IfGB 39/1201 (MIC = 0.17 µM *); *Fusarium proliferatum* IfGB 39/1501 (MIC = 0.17 µM *); *Fusarium sp*. IfGB 39/1101 (MIC = 0.17 µM *); *Aspergillus awamori* ATCC 22,342 (MIC = 0.34 µM *); *F. oxysporum* f.sp. lini IfGB 39/0801 (MIC = 1.38 µM *); *Fusarium bulbigenum* IfGB 39/0301 (MIC = 1.72 µM *); *F. oxysporum* f.sp. vasinfectum IfGB 39/1301 (MIC = 1.72 µM *); *F. solani* IfGB 39/1001 (MIC = 20.67 µM *); *Fusarium poae* IfGB 39/0901 (MIC = 31 µM *); *A. nidulans* DSM 969 (MIC = 34.45 µM *); *A. nidulans* G191 (MIC = 34.45 µM *); *A. giganteus* IfGB 15/0903 (MIC = 68.90 µM *); *A. giganteus* MDH 18,894 (MIC = 68.90 µM *); *Fusarium aquaeductuum* IfGB 39/0101 (MIC = 68.90 µM *); *F. culmorum* IfGB 39/0403 (MIC = 68.90 µM *)	[119]
PAFB	LSKFGGECSLKHNTCTYLKGGKNHVVNCGSAANKKCKSDRHHCEYDEHHKRVDCQTPV (D0EXD3)	58	6500.36/7.74	*Penicillium chrysogenum* (*Trichocomaceae*)	*A. fumigatus* (MIC = 0.25 µM); *A. niger* (MIC = 0.50 µM); *Aspergillus terreus* (MIC = 1 µM)	[120]
PAF	AKYTGKCTKSKNECKYKNDAGKDTFIKCPKFDNKKCTKDNNKCTVDTYNNAVDCD (Q01701)	55	6250.08/7.89		*A. fumigatus* (MIC = 1 µM); *A. niger* (MIC = 0.25 µM); *A. terreus* (MIC = 32 µM)	
AnAFP	LSKYGGECSVEHNTCTYLKGGKDHIVSCPSAANLRCKTERHHCEYDEHHKTVDCQTPV (A2QM98)	58	6517.29/6.24	*A. niger* (*Trichocomaceae*)	*A. flavus* KCTC 1375 (MIC = 8 µM); *A. fumigatus* KCTC 6145 (MIC = (4–8) µM); *F. oxysporum* KCTC 6076 (MIC = (8–15) µM); *F. solani* KCTC 6326 (MIC = 8 µM)	[121]
AcAFP	ATYDGCKCYKKDNICKYKAQSGKT (D3Y2M3)	24	2717.14/8.43	*Aspergillus clavatus* (*Trichocomaceae*)	*F. oxysporum* (MIC = 8.57 µM *); *F. oxysporum* (IC50 = 1.25 µM *)	[122]
NFAP	LEYKGECFTKDNTCKYKIDGKTYLAKCPSAANTKCEKDGNKCTYDSYNRKVKCDFRH (A1D8H8)	57	6625.56/7.92	*Neosartorya fischeri* (*Trichocomaceae*)	*A. niger* (MIC = (3.77–15.09) µM *); *A. nidulans* (MIC = 30.19 µM *)	[123]

The literature, from which the data presented were compiled, was selected from papers published from 1990 to 2021, using the search engines PubMed and ResearchGate, with different associations of the keywords “defensin”, “antifungal” and “plant”. Defensins were classified into four groups based on their organism of origin: the vertebrates, the invertebrates, the plants and the fungi. The minimum inhibitory concentration (MICs) and half-maximal inhibitory concentrations (IC50) noted “*” were calculated from the mass concentrations and molecular mass. The molecular mass (MM) was calculated as the average mass with peptide 2.0. The iso-electric was calculated with IPC 2.0. The accession number is the reference from Uniprot.

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
