# Peer review of "Use of Defensins to Develop Eco-Friendly Alternatives to Synthetic Fungicides to Control Phytopathogenic Fungi and Their Mycotoxins"

_jof, 2022, doi:10.3390/jof8030229_

Round 1
Reviewer 1 Report
Authors wtote an extensive review on the antifungal character of plant defensins, pointing at their potential application in plant protection from fungal diseases and crop losses.
The manuscript is well organised - firstly authors describe defensins on the molecular level, then focus on their antifungal activity and pros and cons of defensin application.
I feel that the subject is handled comprehensively, this review is very valuable.
My only question is about paragraph 4.1 and 4.2. You described the activity of denfensins towards plasma membrane - 4.1 regarding interactions with membrane components and 4.2 refers to the membrane disruption. I am confused why there are seperate paragraphes - do those interactions not lead to membrane disruption? Or perhaps there are other mechanisms connected with plasma membrane, not necesserily leading to pore formation? If so could you please explain?
Minor issue - I would replace "plant fungi" with "plant infecting fungi" (e.g., line 208)
Author Response
Dear reviewer,
We would like to thank you for your useful comments and suggestions. The manuscript has been carefully corrected according to these comments and an item-by-item response to specific comments has been provided below. The indicated line numbers refer to the line numbers of the revised manuscript. In addition, you will find enclosed the revised version with highlighted corrections in yellow of the manuscript jof-1589309 “Use of defensins to develop eco-friendly alternatives to synthetic fungicides to control phytopathogenic fungi and their mycotoxins”.
Authors wrote an extensive review on the antifungal character of plant defensins, pointing at their potential application in plant protection from fungal diseases and crop losses.
The manuscript is well organised - firstly authors describe defensins on the molecular level, then focus on their antifungal activity and pros and cons of defensin application.
I feel that the subject is handled comprehensively, this review is very valuable.
My only question is about paragraph 4.1 and 4.2. You described the activity of denfensins towards plasma membrane - 4.1 regarding interactions with membrane components and 4.2 refers to the membrane disruption. I am confused why there are separate paragraphes - do those interactions not lead to membrane disruption? Or perhaps there are other mechanisms connected with plasma membrane, not necessarily leading to pore formation? If so could you please explain?
We totally agree with this comment and have merged the two paragraphs in the revised version into a new paragraph entitled “Interactions with host membrane components and induction of fungal membranes disorders”. Indeed, while not all defensins induce pore formation, they all induce membranes disorders when they interact with membranes components.
Minor issue - I would replace "plant fungi" with "plant infecting fungi" (e.g., line 208)
This has been corrected in the revised version (P2 line 62 ; P5 L201 ; P18 L207, 214).
Reviewer 2 Report
The authors are proposing for publication to the Journal of Fungi the review “Use of defensins to develop eco-friendly alternatives to synthetic fungicides to control phytopathogenic fungi and their mycotoxins”. The review is based on a description of the classification of the defensins with an important list of molecules proposed to satisfy the curiosity of the readers: The authors have subdivided the review in six different sections from an Introduction to fix the reader on the state of the art on what is recognized as the most widespread class of antimicrobial peptides (AMPs) among the animal and plant kingdoms. Following the Introduction, the authors are presenting the Origin and structural characteristics of this class of AMPs, before entering the activities recorded so far for the defensins against fungal phytopathogens. Then the authors are entering in a discussion on a rather moving and controversial topic regarding AMPs, precisely their mechanisms of action, which remains an open question. Finally, the authors prior their conclusions are proposing an interesting discussion on the opportunities to make these AMPs defensins alternatives to synthetic fungicides to kill fungi and/or to neutralize mycotoxins that may represent toxic and detrimental food contaminant for human and livestock. The last section 5.2 is really well documented and attractive.
The review is supported by a quite exhaustive list of references (reference 105 needs to be adjusted) which needs to be adjusted according to the suggestion to add more than one review (148) while the authors are suggesting to the reader “to consult relevant reviews (page 17, line 351) and to include the publication proposed by Aumer et al on the mechanisms of action of a recombinant antifungal defensins on the phytopathogenic fungus Botrytis cinerea, which will be appropriate in section 4, subsection 4.1.
Concerning section 2 on the origin of the defensins. Page 3, line 120. Line 120 same page what is the meaning of “a conserved structure”? this is not the case for the defensins as the structures are really diverse both at the primary structure, according to the cysteine pairing and according to their 3D when known. This should be clarified and adjusted. Line 122, the authors are proposing that defensins are “small proteins of 12 to 50 amino acids”.
Can the authors confirm the existence of a 12 residue defensin? Lines 125-126, this sentence is really confusing. “defensins share a common fold…, this is not the case. This should be clarified to avoid any confusion, even if the topic is more on defensins with antifungal properties, it is suggested to mention the big-defensins and the teta defensins. Page 4, line 161. Do the authors include the question of susceptibility to proteolysis? This point was raised later in the manuscript (page 22, line 556…).
Page 5, section “Most defensins...[56], the message is not clear due to redundancies in the writing. It is suggested to revised this paragraph. Regarding the discussion on the precursor of defensins it is suggested to discuss the question of the presence of the pro-domain, its physico chemical properties (acidic) and its function as neutralizer of the cationicity of the mature defensins.
Page 14, lines 218-221. The sentence is confusing regarding the “six defensins from vertebrates”. What is the message of the authors? There are more than 6 vertebrate defensins in the literature. Same page. When the authors are mentioning the activity of defensins against fungi. The reviewer agrees with what is mentioned by the authors (morer antibacterial defensins than antifungal defensins) but this should be placed in its historical context. Historically, researchers where focused on antibacterial properties and not on antifungal activities. This type of activity was evaluated thanks to the plant defensin discovery and tests were less popular in laboratories than antibacterial assays. In fact, antifungal properties for defensins is underestimated.
Some minor comments: page 1, line 31, in vitro (not invitro). Choose the table writing to Table or table within the manuscript. Page 3 line 114, I would write Firstly rather than firs as you have Secondly. Also, sometime the name of the fungal strains, plants etc needs to be as a full name (e.g. page 17, line 340, Pachyrhizus erosus instead of P. erosus).
Page 16, lines 314 and 317, the short domains mentioned are interesting but an information on their location within the full sequence would be more interesting precisely if the 3D structures of these defensins are available. This is an information interesting when you are performing SAR studies to optimize a molecule. Line 320 change for Threonine and Phenylalanine to keep homogeneity. Line 328, the authors are mentioning “does not affect fungal membrane integrity [143]”. Can the authors add information on this? The discussion on the transgenic plants expressing defensins is interesting. However, the authors are not mentioning any data on the stability of the transgene from generation to generation, on its effect on plant growth and on the impact on the nutritional value of the plant expressing such peptides. Is there any data available that the authors can discussed?
In Table 1 what is the meaning of X as first amino acid in some peptide sequences proposed? Also in Table 1, the headings of the columns organism must be Organism, instead of MW MM would be more appropriate. Regarding this Table, a suggestion would be to split it in two different tables. The first one providing all sequences in 1 or 2 lines and not as it is, to add, plus the accession numbers a useful link for the reader to retrieve the original sequences, the MM with a mention on how this value was calculates (average or monoisotopic mass, do this value include the removal of the 1Da for each cysteine engaged in a pairing, presence or not of an amidation at the C-terminus), an additional column with the isoelectric point and another one proposing the number of amino acids. The second table may propose the biological properties recorded for the listed peptides to simplify the reading rather hard in Table 1. In figure 1, it is recommended to add the two disulfide bridges in the gomesin 3D structure proposed. There few Latin names to write in italic.
Author Response
Dear reviewer,
We would like to thank you for your useful comments and suggestions. The manuscript has been carefully corrected according to these comments and an item-by-item response to specific comments has been provided below. The indicated line numbers refer to the line numbers of the revised manuscript. In addition, you will find enclosed the revised version with highlighted corrections in yellow of the manuscript jof-1589309 “Use of defensins to develop eco-friendly alternatives to synthetic fungicides to control phytopathogenic fungi and their mycotoxins”.
The authors are proposing for publication to the Journal of Fungi the review “Use of defensins to develop eco-friendly alternatives to synthetic fungicides to control phytopathogenic fungi and their mycotoxins”. The review is based on a description of the classification of the defensins with an important list of molecules proposed to satisfy the curiosity of the readers: The authors have subdivided the review in six different sections from an Introduction to fix the reader on the state of the art on what is recognized as the most widespread class of antimicrobial peptides (AMPs) among the animal and plant kingdoms. Following the Introduction, the authors are presenting the Origin and structural characteristics of this class of AMPs, before entering the activities recorded so far for the defensins against fungal phytopathogens. Then the authors are entering in a discussion on a rather moving and controversial topic regarding AMPs, precisely their mechanisms of action, which remains an open question. Finally, the authors prior their conclusions are proposing an interesting discussion on the opportunities to make these AMPs defensins alternatives to synthetic fungicides to kill fungi and/or to neutralize mycotoxins that may represent toxic and detrimental food contaminant for human and livestock. The last section 5.2 is really well documented and attractive.
The review is supported by a quite exhaustive list of references (reference 105 needs to be adjusted) which needs to be adjusted according to the suggestion to add more than one review (148) while the authors are suggesting to the reader “to consult relevant reviews (page 17, line 351) and to include the publication proposed by Aumer et al on the mechanisms of action of a recombinant antifungal defensins on the phytopathogenic fungus Botrytis cinerea, which will be appropriate in section 4, subsection 4.1.
The reference 105 (109 in the revised version) has been corrected such as the reference 104 (108 in the revised version).
P17, line 351 (P 21, Line 363 in the revised version): Thanks for this well taken comment; the sentence has been modified accordingly.
The recent publication of Aumer et al. has been included and discussed in the revised version of our manuscript. (P 19, Line 293 in the revised version)
Concerning section 2 on the origin of the defensins. Page 3, line 120. Line 120 same page what is the meaning of “a conserved structure”? this is not the case for the defensins as the structures are really diverse both at the primary structure, according to the cysteine pairing and according to their 3D when known. This should be clarified and adjusted. Line 122, the authors are proposing that defensins are “small proteins of 12 to 50 amino acids”. Can the authors confirm the existence of a 12 residue defensin?
According to the previous comments, we have modified the sentence containing the expression “conserved structure” (P3, Line 122 in the revised version). We also modified the “12 to 50 amino acids” range (which actually is commonly associated with AMPs and is inaccurate for defensins).
Lines 125-126, this sentence is really confusing. “defensins share a common fold…, this is not the case. This should be clarified to avoid any confusion, even if the topic is more on defensins with antifungal properties, it is suggested to mention the big-defensins and the teta defensins.
The sentence (L125-126 / L128-129 in the revised version) has been modified and a paragraph mentioning big- and θ-defensins included (P 4 L 142-145).
Page 4, line 161. Do the authors include the question of susceptibility to proteolysis? This point was raised later in the manuscript (page 22, line 556…).
A comment dealing with protection against proteolysis conferred by disulfide bridges has been included in the revised version (P 4 L 169 ).
Page 5, section “Most defensins...[56], the message is not clear due to redundancies in the writing. It is suggested to revised this paragraph. Regarding the discussion on the precursor of defensins it is suggested to discuss the question of the presence of the pro-domain, its physico chemical properties (acidic) and its function as neutralizer of the cationicity of the mature defensins.
We totally agree with the fact that this paragraph was unclear. It has been simplified and a comment regarding the role of the pro-domain (related to its physicochemical properties) has been added (p 5 , L 186-195).
Page 14, lines 218-221. The sentence is confusing regarding the “six defensins from vertebrates”. What is the message of the authors? There are more than 6 vertebrate defensins in the literature.
This sentence relates to defensins with a reported bioactivity against phytopathogenic fungi. To avoid confusion, the previous clarification has been added (P18, L219).
Same page. When the authors are mentioning the activity of defensins against fungi. The reviewer agrees with what is mentioned by the authors (more antibacterial defensins than antifungal defensins) but this should be placed in its historical context. Historically, researchers where focused on antibacterial properties and not on antifungal activities. This type of activity was evaluated thanks to the plant defensin discovery and tests were less popular in laboratories than antibacterial assays. In fact, antifungal properties for defensins is underestimated.
We totally agree with this comment and a sentence has been included in the revised version with the aim to qualify the statement that plant defensins exhibit activity primarily against fungi while fungal and animal defensins have efficient antibacterial properties. (P19, L242-247)
Some minor comments: page 1, line 31, in vitro (not invitro). Choose the table writing to Table or table within the manuscript. Page 3 line 114, I would write Firstly rather than firs as you have Secondly. Also, sometime the name of the fungal strains, plants etc needs to be as a full name (e.g. page 17, line 340, Pachyrhizus erosus instead of P. erosus).
The manuscript has been corrected (in vitro, firstly, Table instead of table). We have also reviewed all the manuscript (hoping we have not omitted some errors) to check the names (full name or abbreviation) of plant, animal, fungal species. Regarding P. erosus (p17, line340), the full name was first mentioned in Table 1.
Page 16, lines 314 and 317, the short domains mentioned are interesting but an information on their location within the full sequence would be more interesting precisely if the 3D structures of these defensins are available. This is an information interesting when you are performing SAR studies to optimize a molecule.
This information has been added (p 20, L328)
Line 320 change for Threonine and Phenylalanine to keep homogeneity.
This has been done.
Line 328, the authors are mentioning “does not affect fungal membrane integrity [143]”. Can the authors add information on this?
The sentence has been completed and a paragraph included (p 21, L339-343)
The discussion on the transgenic plants expressing defensins is interesting. However, the authors are not mentioning any data on the stability of the transgene from generation to generation, on its effect on plant growth and on the impact on the nutritional value of the plant expressing such peptides. Is there any data available that the authors can discussed?
At the end of section 5.1, there is a paragraph gathering published data that addressed the negative impact induced by defensin transgene introduction on various plant traits. (p25, L498-503)
In Table 1 what is the meaning of X as first amino acid in some peptide sequences proposed? Also in Table 1, the headings of the columns organism must be Organism, instead of MW MM would be more appropriate. Regarding this Table, a suggestion would be to split it in two different tables. The first one providing all sequences in 1 or 2 lines and not as it is, to add, plus the accession numbers a useful link for the reader to retrieve the original sequences, the MM with a mention on how this value was calculates (average or monoisotopic mass, do this value include the removal of the 1Da for each cysteine engaged in a pairing, presence or not of an amidation at the C-terminus), an additional column with the isoelectric point and another one proposing the number of amino acids. The second table may propose the biological properties recorded for the listed peptides to simplify the reading rather hard in Table 1.
In Table 1, the X were errors corrected in this version. Since our review is mainly focusing on the biological activity of defensins, we preferred to not split the table 1 in two tables with one only dedicated to physicochemical properties. Nonetheless, we totally agree that this table must be improved and completed with some important information. In the revised version, the table has been presented in a landscape format so that the sequences are reported in only 1-2 lines. As recommended by the reviewer, the accession number from Uniprot, pi and number of amino acids have been added such as clarification regarding the MM (MM instead of MW has been used) and pi calculations.
In figure 1, it is recommended to add the two disulfide bridges in the gomesin 3D structure proposed.
This has been added.
There few Latin names to write in italic.
This has been checked throughout the manuscript.